# FiRL : Finslerian Reinforcement Learning for Risk-Aware Anisotropic Locomotion

## Abstract

Legged locomotion is inherently *anisotropic* and risk-sensitive: the energy cost and risk of failure vary significantly with the direction and speed of motion. Standard reinforcement learning (RL) methods neglect this asymmetry, typically using isotropic cost/reward functions and optimizing only for expected returns. This leaves agents vulnerable to rare but catastrophic outcomes. We propose **Finslerian Reinforcement Learning (FiRL)**, a novel RL framework that integrates a *Finsler metric* into the cost function for directional energy-awareness, and optimizes a Conditional Value-at-Risk (CVaR$_\alpha$) objective for tail-risk robustness. FiRL formulates the locomotion cost as $F(x, v)$, a Finsler metric that varies with state $x$ and motion $v$, capturing uphill vs. downhill effort, lateral friction, and other direction-dependent costs. We derive a risk-sensitive Bellman equation based on CVaR and prove that the corresponding CVaR–Finsler Bellman operator is a $\gamma$-contraction, yielding a unique fixed-point value function that induces a *quasi-metric* structure (satisfying a triangle inequality despite asymmetry). We develop a FiRL actor–critic algorithm to learn policies under this anisotropic, risk-averse objective. In simulated *MuJoCo and IsaacSim* locomotion benchmarks, FiRL achieves safer and more energy-efficient behaviors than SOTA baselines (e.g., risk-neutral PPO). For example, on a $12°$ slope Hopper task FiRL reduces worst-case (CVaR$_{0.1}$) impact forces by over 35% and total energy cost by 15%, while attaining a higher success rate.

## 1 Introduction

Legged robots offer unique advantages in navigating unstructured terrains, but they also face significant challenges in energy management and safety Miki et al. (2022). Recent advances in reinforcement learning (RL) have produced agile locomotion controllers for complex robots Hwangbo et al. (2019); Rudin et al. (2022). However, standard RL formulations often assume *isotropic* (direction-agnostic) cost or reward functions and optimize solely for expected return Rudin et al. (2022); Hwangbo et al. (2019). This overlooks two crucial aspects of real-world locomotion: (1) **Directional anisotropy in cost** – moving in different directions can incur vastly different energy expenditures and wear Fu et al. (2022)(e.g., climbing uphill requires far more work than going downhill, sharp turns can waste momentum, descending steps quickly can be dangerous); and (2) **Tail-risk sensitivity** – rare but high-cost events such as slipping, falling, or hardware strain must be minimized for safe deployment. By ignoring anisotropic effort and focusing only on average performance, conventional RL policies may appear efficient on flat terrain but perform poorly on slopes or under disturbances, suffering catastrophic failures in the worst cases Shi et al. (2024); Du et al. (2022).

Recent work on geometry-aware RL has explored embedding inductive biases about distance and dynamics into value functions. For example, Wang et al. (2023) enforce a *quasi-metric* structure on value estimates to satisfy triangle inequality and asymmetry, improving generalization in goal-reaching tasks. Others have applied *Riemannian* geometry to RL, e.g. shaping policy updates or value functions using symmetric metrics Abbasi-Yadkori & Mahdavi (2022); Kan et al. (2021); Ravindran et al. (2023). These approaches demonstrate the benefit of geometric priors, but they either assume simplified settings (constant speeds, discrete goals in quasi-metric RL) or impose symmetric structures that cannot capture direction-specific costs. Meanwhile, in risk-sensitive and safe RL, methods like CVaR optimization Chow et al. (2015); Tamar et al. (2015) and distributional RL bias the training towards safer outcomes by focusing on the worst-case returns. Notably, Schneider et al.

(2024) apply distributional RL to quadrupedal locomotion, showing that risk-aware training can yield cautious gait policies on a real robot. While earlier methods have explored reward shaping for anisotropy and CVaR-based risk objectives separately, the integration of direction-dependent costs with explicit tail-risk optimization has not been addressed.

In this paper, we propose **Finslerian Reinforcement Learning (FiRL)**, a novel framework that integrates differential geometry with risk-sensitive RL for legged locomotion. FiRL introduces a *Finsler metric* into the cost function, allowing the agent to account for direction-dependent effort: for instance, uphill moves incur higher instantaneous cost than downhill moves, and lateral motions incur frictional penalties (Fig. 1). Simultaneously, FiRL optimizes a *Conditional Value-at-Risk (CVaR)* objective, which biases learning towards minimizing the worst-case outcomes (the tail of the cost distribution) rather than just the mean. By integrating these components, FiRL produces policies that proactively avoid energetically costly maneuvers *and* reduce the probability of catastrophic failures.

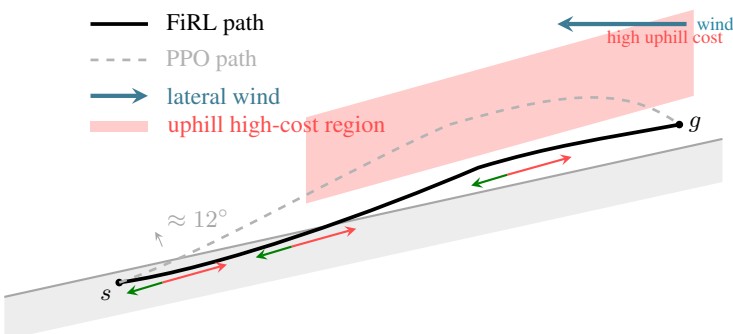

Figure 1: : **FiRL vs. PPO on a slope with lateral wind.** The Finsler cost induces higher uphill cost (red wedge), a lateral wind biases motion, and FiRL (solid) selects a safer, lower-cost route than PPO (dashed). The table contrasts risk-neutral PPO with FiRL's risk-aware objectives (high- and low-level updates)

**Our main contributions**:

- **Finsler Cost with Tail-Risk Objective**: We propose a new per-step cost formulation using a Finsler metric $F(x, v)$ that depends on the agent's state $x$ and motion direction $v$. This formulation explicitly embeds anisotropic energetics into the cost (rather than treating it as a heuristic), and we optimize a CVaR (Conditional Value-at-Risk) criterion of the cumulative cost. By integrating anisotropic geometry directly into the cost model along with a tail-risk objective, our approach can capture direction-dependent energy penalties in a principled way.

- **Risk-Sensitive Bellman Operator & Induced Geometry**: We derive a CVaR–Finsler Bellman operator for our risk-sensitive setting and prove that it is a $\gamma$-contraction under mild boundedness conditions. This theoretical result ensures the existence of a unique fixed point (optimal value function) and standard convergence guarantees for dynamic programming. Moreover, we show that the optimal value function under this Finsler–CVaR framework induces a quasi-metric on the state space (satisfying the triangle inequality but not symmetry), providing a novel geometric interpretation of directional costs in the value landscape.

- **FiRL–AC Algorithm (Actor–Critic)**: We develop FiRL–AC, a new actor–critic learning algorithm that incorporates the anisotropic Finsler cost and CVaR objective into practical reinforcement learning. In FiRL–AC, the per-step cost uses $F(x, v)$ to penalize movements directionally, and the critic is trained with CVaR-based targets to focus on tail outcomes. To rigorously evaluate the impact of anisotropic cost vs. standard cost and CVaR vs. expected cost, we perform a controlled 2×2 ablation study with matched hyperparameters.

- **Empirical Results on Anisotropic Terrains**: Through extensive experiments on challenging locomotion tasks (e.g., sloped ground and lateral wind), we show that FiRL achieves a superior trade-off between energy efficiency and safety (risk of high-impact events) compared to baseline approaches. FiRL consistently shifts the energy–risk Pareto frontier:

it reduces worst-case costs (lower CVaR0.1 of energy usage and impact forces) and limits peak impact forces, while maintaining or decreasing average energy consumption. Notably, on a steep $12°$ slope Hopper task, FiRL lowers the CVaR0.1 of ground-impact force by over 35% and the total energy expended by about 15% relative to the risk-neutral baseline, all while increasing the task success rate.

## 2 RELATED WORK

**Geometry in Reinforcement Learning:** Incorporating geometric structure into RL has shown promise in improving generalization and sample efficiency. Wang et al. (2023) introduced *quasi-metric RL* (QRL), enforcing value functions to satisfy the triangle inequality and asymmetry properties of a distance function. By shaping the learned value as a distance-to-goal, QRL improved goal-directed navigation. However, QRL assumes fixed-speed trajectories and discrete goal states, limiting its applicability to continuous locomotion tasks with varying speeds. Other works have explored *Riemannian* metrics in RL: for example, Abbasi-Yadkori & Mahdavi (2022) and Kan et al. (2021) modify policy optimization by computing gradients with respect to a Riemannian manifold (such as one defined by state covariance), and Ravindran et al. (2023) shape value functions using manifold distances to improve generalization. These methods impose symmetric metrics (distance is the same in all directions) and do not address directional asymmetry in costs. In contrast, *Finsler geometry* generalizes Riemannian geometry by allowing the metric to be asymmetric and state-dependent; our work is, to our knowledge, the first to leverage Finsler metrics in RL to capture anisotropic effort.

**Locomotion and energy shaping.** Energy–efficient and safe locomotion have long been pursued in robotics via engineered objective terms (e.g., squared torques, contact and velocity penalties) and safety layers grounded in control theory. Energy/power shaping is classical in passivity-based control of mechanical systems, while safety-critical control frequently employs *control barrier functions* to filter actions through real-time QPs (Ames et al., 2016; 2019). In modern benchmarks, MuJoCo locomotion tasks pair forward-progress rewards with isotropic $\ell_2$ control and contact costs, without explicit direction-dependent running costs (Tassa et al., 2018); consequently, standard RL setups optimize expected return under largely direction-agnostic shaping, even as learned policies reach strong hardware performance (Hwangbo et al., 2019; Rudin et al., 2022). Recent work has begun to target energy explicitly in legged locomotion with task-specific regularizers or reward terms (Liang et al., 2025), but these typically remain symmetric in motion direction. Our approach provides a principled way to encode domain knowledge such as "uphill motion is costly" through a Finsler metric $F(x, v)$—a state- and direction-dependent cost rooted in differential geometry (Bao et al., 2000; Ratliff et al., 2021)—rather than ad-hoc reward tweaks, yielding anisotropic and asymmetric effort models aligned with locomotion physics. We discuss additional related work in Appendix B.

## 3 METHODOLOGY

### 3.1 PROBLEM FORMULATION AND PRELIMINARIES

We consider a standard Markov Decision Process (MDP) formalism for the locomotion task: states $x \in \mathcal{X}$, actions $u \in \mathcal{U}$, transition dynamics $x' = f(x, u)$ (possibly stochastic), and a discount factor $\gamma \in [0, 1)$. However, instead of a conventional scalar reward, we define a *state-action cost* via a Finsler metric $F(x, v)$, where $v$ represents the *motion vector* (e.g., the instantaneous velocity) induced by taking action $u$ in state $x$. Intuitively, $F(x, v)$ measures the **effort or "distance"** incurred by moving with velocity $v$ at state $x$; it generalizes the notion of energy expenditure or risk cost for a small step. We first introduce the components of our Finslerian cost function for locomotion. In our design (Fig. 2), $F(x, v)$ comprises three terms capturing different aspects of motion cost:

**(1) Kinetic energy term $F_{\text{energy}}$:** This term accounts for the basic dynamic effort of movement. We define it as

$$F_{\text{energy}}(x, v) = \sqrt{v^\top M(x)\, v} \,, \tag{1}$$

where $M(x)$ is a positive-definite weight (or inertia) matrix that can vary with state. In our design, $M(x)$ assigns larger weights to motion along certain axes that are energetically expensive (for example, vertical movements against gravity, or rotations of the body orientation). $F_{\text{energy}}$ is symmetric

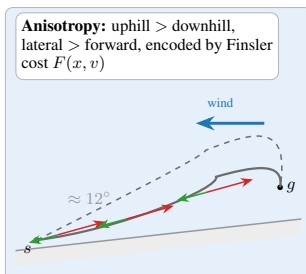 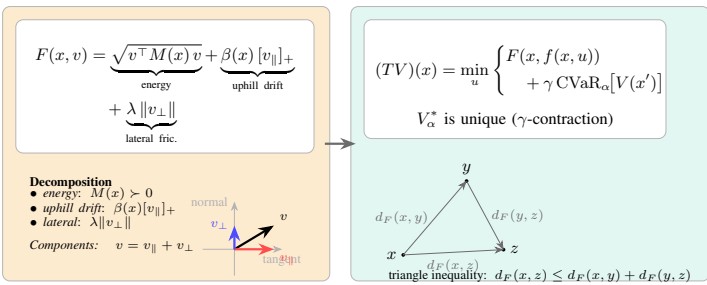

Figure 2: **FiRL Overview. Left:** Anisotropic terrain (slope + wind) induces direction-dependent effort: uphill and lateral motions are costly, downhill cheaper. **Middle:** The local Finsler cost $F(x, v)$ decomposes into energy, uphill drift, and lateral friction terms. **Right:** The CVaR Bellman operator combines $F$ with the tail-risk of future value, yielding a unique fixed point $V_\alpha^*$ whose induced path cost $d_F$ satisfies a triangle inequality (quasi-metric).

in $v$ (since it depends on $v$ quadratically) and ensures that faster motions or motions in "heavier" directions yield higher cost.

**(2) Uphill drift term $F_{\text{drift}}$:** To impose an asymmetry between upward (against gravity) and downward movements, we include a drift term:

$$F_{\text{drift}}(x, v) = \beta(x) \max(0, v_\parallel), \tag{2}$$

where $v_\parallel$ denotes the component of $v$ in the direction of gravity (i.e., vertical velocity) and $\beta(x) \geq 0$ is a state-dependent scalar weight that increases with the incline of the terrain at $x$. The $\max(0, \cdot)$ ensures that only motion *against gravity* (upward) incurs this additional cost; if $v_\parallel < 0$ (downward movement), $F_{\text{drift}}$ contributes zero. Thus, $\beta(x)$ effectively scales the extra effort required to climb: on steeper slopes, $\beta(x)$ is higher, meaning uphill steps are much more costly, whereas moving downward or on flat ground yields no drift penalty. This term makes $F$ *asymmetric*, since $F(x, v) \neq F(x, -v)$ in general (climbing up vs. down yields different costs).

**(3) Lateral friction term $F_{\text{friction}}$:** Legged robots also experience higher risks and energy loss during lateral or non-straight movements (e.g., side-stepping, turning sharply can cause skidding or inefficient gaits). We capture this via:

$$F_{\text{friction}}(x, v) = \lambda \|v_\perp\|, \tag{3}$$

where $v_\perp$ is the component of $v$ orthogonal to the robot's primary forward heading direction, and $\lambda > 0$ is a constant coefficient. This term is linear in the lateral speed magnitude, penalizing any sideways motion. $F_{\text{friction}}$ breaks isotropy by distinguishing forward vs. lateral movement directions, although it is symmetric with respect to left vs. right ($v_\perp$ enters through its norm). Essentially, $F_{\text{friction}}$ adds a cost for curving or side-slipping, discouraging high-curvature trajectories that could lead to loss of traction.

**Total Finsler Metric:** We combine the above components into the total cost metric as a weighted sum:

$$F(x, v) = w_e F_{\text{energy}}(x, v) + w_d F_{\text{drift}}(x, v) + w_f F_{\text{friction}}(x, v), \tag{4}$$

with positive weights $w_e, w_d, w_f > 0$ chosen to balance the contributions of each term.[1]

**Definition 3.1** (Finsler Metric). *For each fixed state $x$, the function $v \mapsto F(x, v)$ defined by equation 4 is a* Finsler metric *on the tangent space $T_x\mathcal{X}$. It satisfies: (i) $F(x, v) \geq 0$ for all $v$, and $F(x, v) = 0$ if and only if $v = \mathbf{0}$; (ii) positive homogeneity: $F(x, \lambda v) = \lambda F(x, v)$ for all $\lambda > 0$; (iii) smoothness in $v$ (except potentially at $v = \mathbf{0}$ or points of nondifferentiability due to the max, which can be handled by subgradients). Unlike Riemannian metrics, a Finsler metric need not be symmetric: in general $F(x, v) \neq F(x, -v)$, which allows encoding one-way directional differences (e.g., uphill vs. downhill).*

---

[1]We selected these weights via a hyper-parameter search to achieve a reasonable balance; see Appendix M for the chosen values. For simplicity, one can set $w_e = w_d = w_f = 1$ if units are commensurate.

Our chosen $F(x, v)$ indeed satisfies these properties: $F(x, v) > 0$ for $v \neq 0$, it is positively homogeneous (each component is homogeneous of degree 1 in $v$), and it is smooth except where $v_\parallel = 0$ (the 'kink' at the switch between uphill and downhill; $\beta(x)$ can be made differentiable and we handle the subgradient at $v_\parallel = 0$ in practice). Importantly, $F$ is asymmetric due to the drift term, but this asymmetry encodes meaningful physical bias (gravity's effect). By construction, integrating $F$ along a trajectory yields a *path cost*:

$$d_F(\tau) = \int_0^1 F(\tau(t), \dot{\tau}(t)) \, dt,$$

for any path $\tau : [0, 1] \to \mathcal{X}$. In Appendix E, we show that $d_F$ defines a valid *quasi-metric* distance on $\mathcal{X}$: it obeys the triangle inequality (so costs are path-consistent) while allowing $d_F(x, y) \neq d_F(y, x)$. This quasi-metric perspective will be revisited in our theoretical results.

**CVaR risk objective.** In the FiRL framework, the agent's goal is to minimize not just the expected cumulative cost, but the *Conditional Value-at-Risk (CVaR)* of the cumulative cost distribution. Formally, let $\tau$ denote a trajectory and $J(\tau) = \sum_{t=0}^{T-1} \gamma^t F(x_t, v_t)$ be the total discounted cost (with horizon $T$ which could be infinite for continuing tasks). For a given risk level $\alpha \in (0, 1]$, the $\text{CVaR}_\alpha$ of the return is defined as

$$\text{CVaR}_\alpha(J) = \mathbb{E}\big[J(\tau) \,\big|\, J(\tau) \text{ is in the worst } \alpha^{-1}\text{-quantile}\big],$$

i.e., the average cost of the worst $\alpha$ fraction of episodes. Intuitively, smaller $\alpha$ focuses more on the absolute worst cases. Minimizing $\text{CVaR}_\alpha$ encourages the policy to avoid catastrophic tails at the possible expense of some optimality in the average case (when $\alpha = 1$, CVaR becomes the expectation, recovering the risk-neutral objective). We denote by $V_\alpha(x)$ the risk-sensitive value function: $V_\alpha(x) = \min_\pi \text{CVaR}_\alpha^\pi(J \,|\, x_0 = x)$, the minimal CVaR of total cost starting from state $x$ under an optimal policy $\pi$. The corresponding optimal policy aims to minimize the tail risk of the return.

# 4 FINSLERIAN CVAR BELLMAN EQUATION AND FIRL-AC ALGORITHM

## 4.1 CVAR BELLMAN OPERATOR AND CONTRACTION

To solve for the optimal risk-sensitive value, we extend the Bellman equation to the CVaR setting with our anisotropic cost. Define the *risk-aware Bellman operator $T$* acting on any candidate value function $V : \mathcal{X} \to \mathbb{R}$ as:

$$(TV)(x) = \min_{u \in \mathcal{U}} \left\{ F(x, f(x, u)) + \gamma \, \rho_\alpha\big[V(x')\big] \right\}, \tag{5}$$

where $x' \sim P(\cdot \,|\, x, u)$ is the next state, and $\rho_\alpha[V(x')] = \text{CVaR}_\alpha(V(x'))$ denotes the CVaR (with respect to the randomness in $x'$) of the next-state value. Intuitively, $TV(x)$ computes the immediate cost $F(x, f(x, u))$ for a chosen action $u$, plus the discounted $\alpha$-tail of future costs if we act optimally thereafter. The optimal value function $V_\alpha^*(x)$ should satisfy the fixed-point equation $V_\alpha^*(x) = (TV_\alpha^*)(x)$ for all $x$. However, directly computing this fixed point is challenging due to the minimization and the CVaR nested inside. Instead, we will show that $T$ is a contraction mapping, which means value iteration or other iterative algorithms can converge to $V_\alpha^*$.

We leverage properties of CVaR (a *coherent risk measure*) to establish contraction. CVaR has two key properties: (i) *Monotonicity* – if $V_1(x') \geq V_2(x')$ for all outcomes $x'$, then $\rho_\alpha[V_1(x')] \geq \rho_\alpha[V_2(x')]$; (ii) *Positive homogeneity* – $\rho_\alpha[c \, X] = c \, \rho_\alpha[X]$ for $c \geq 0$. Given these, one can show that $T$ is a $\gamma$-contraction in the $\ell_\infty$ norm, similar to the standard expected Bellman operator.

**Theorem 4.1** (CVaR–Finsler Bellman Contraction). *Assume the transition dynamics yield bounded costs and that $F(x, v)$ is bounded and Lipschitz in $x$ (ensuring $TV$ maps bounded continuous functions to bounded continuous functions). Then $T$ as defined in equation 5 is a contraction mapping with factor $\gamma < 1$ in the sup norm. That is, for any two value functions $V_1, V_2$,*

$$\|TV_1 - TV_2\|_\infty \leq \gamma \, \|V_1 - V_2\|_\infty.$$

*Consequently, $T$ has a unique fixed point $V_\alpha^*$, and iterative application $V^{(k+1)} \leftarrow TV^{(k)}$ converges to $V_\alpha^*$ from any initial $V^{(0)}$. Moreover, $V_\alpha^*$ is the optimal risk-sensitive value function for the MDP (the CVaR-optimal value).*

*Proof.* The proof (detailed in Appendix E) adapts standard contraction arguments to the CVaR case. For any two value functions $V_1, V_2$ and any state $x$, using the definition equation 5 we get:

$$(TV_1)(x) - (TV_2)(x) = \min_u\{F(x, f(x,u)) + \gamma\,\rho_\alpha[V_1(x')]\} - \min_u\{F(x, f(x,u)) + \gamma\,\rho_\alpha[V_2(x')]\}$$

$$\leq \max_u \gamma\left(\rho_\alpha[V_1(x')] - \rho_\alpha[V_2(x')]\right),$$

since the immediate cost terms cancel for a given action. By CVaR's monotonicity, $\rho_\alpha[V_1(x')] - \rho_\alpha[V_2(x')] \leq \rho_\alpha[V_1(x') - V_2(x')] \leq \|V_1 - V_2\|_\infty$. Thus $(TV_1)(x) - (TV_2)(x) \leq \gamma\|V_1 - V_2\|_\infty$. Reversing the roles of $V_1, V_2$ yields a symmetric bound in the other direction. Therefore $|(TV_1)(x) - (TV_2)(x)| \leq \gamma\|V_1 - V_2\|_\infty$ for all $x$, proving the contraction. Uniqueness of the fixed point and convergence follow by Banach's fixed-point theorem . (Note: This argument requires $\gamma < 1$ and boundedness to ensure $\rho_\alpha$ is well-behaved; these conditions hold in typical discounted-cost settings.) $\square$

**Quasi-Metric Value Property.** The fixed-point value function $V_\alpha^*(x)$ under our Finslerian cost has an interesting structural property: it is *consistent with a quasi-metric*. In informal terms, if we interpret $V_\alpha^*(x)$ as the "distance-to-go" (in terms of cumulative cost) from $x$ to some goal or terminal condition (e.g., episode end with zero cost), then $V_\alpha^*$ obeys a triangle inequality. More formally, we can show (Appendix E) that for any three states $x, y, z$,

$$V_\alpha^*(x \to z) \leq V_\alpha^*(x \to y) + V_\alpha^*(y \to z),$$

where $V_\alpha^*(x \to y)$ denotes the optimal cost-to-go from $x$ to reach $y$ (this can be rigorously defined in an episodic setting or by treating $y$ as a waypoint). This property stems from the fact that the instantaneous cost between states, given by the Finsler metric path integral, satisfies the triangle inequality, and the CVaR optimization preserves a form of *path consistency* for worst-case costs. In addition, FiRL's optimal value can be seen as inducing a quasi-metric on the state space: it behaves like a distance function (ensuring that detours cannot reduce cost) even though the cost is asymmetric. This is in contrast to standard expected value functions, which need not satisfy any such inequality without special regularization.

**Geometric interpretation: quasi-metric from Finsler + CVaR.** Because $F(x, v)$ can assign different cost to $v$ and $-v$ and the CVaR objective emphasizes tail outcomes, the optimal value function $V_\alpha^*$ induces an asymmetric distance-like quantity. For example, in a simple 1D slope where uphill costs 5 and downhill costs 1, the cost-to-go from bottom to top is about 5 while from top to bottom is about 1: $d(\text{bottom}, \text{top}) \approx 5 \neq 1 \approx d(\text{top}, \text{bottom})$. We show in Appendix E that such distances satisfy a triangle inequality (over concatenated paths), making $V_\alpha^*$ a quasi-metric on states.

## 4.2 FiRL-AC: Finsler Actor–Critic Algorithm

Given the theoretical discussion above, we now describe our learning algorithm, **FiRL-AC**, which is an actor–critic method tailored to the CVaR objective with Finslerian costs. Like standard actor–critic, we maintain two function approximators: a critic $V_\alpha(x; \phi)$ parameterized by $\phi$ to estimate the risk-sensitive value, and an actor (policy) $\pi(u|x; \theta)$ parameterized by $\theta$ to select actions. The training loop alternates between *policy evaluation* (updating $\phi$ to better estimate the CVaR value under the current policy) and *policy improvement* (updating $\theta$ to reduce the CVaR of returns).

**CVaR-Critic update.** Evaluating a CVaR value is more involved than a standard value. We employ a sampling-based approach inspired by distributional RL: for a batch of states, we estimate the CVaR of their returns by simulating multiple trajectories. Specifically, for each state $x_i$ in a batch (collected from the current policy's experience), we sample $K$ trajectories (or rollouts) starting from $x_i$, obtaining $K$ realizations of the return $J_i^{(1)}, \ldots, J_i^{(K)}$. We then approximate $\text{CVaR}_\alpha(J|x_i)$ by taking the average of the worst $\lceil \alpha K \rceil$ returns among those $K$. Denote this empirical CVaR as $\widehat{\rho}_\alpha[J|x_i]$. Using one-step transition samples, we then construct a *Bellman target* for $V_\alpha(x_i)$:

$$y_i = F\big(x_i,\, f(x_i, u_i)\big) + \gamma\,\widehat{\rho}_\alpha\Big[V_\alpha\big(x_{i+1}; \phi^-\big)\Big], \tag{6}$$

where $u_i$ is the action taken in $x_i$ (by the current policy), $x_{i+1}$ is the resulting next state, and $V_\alpha(x_{i+1}; \phi^-)$ is the critic's estimate of the next-state value (we may use a target network or previous

iteration parameters $\phi^-$ to stabilize training, as in DQN). The term $\widehat{\rho}_\alpha[V_\alpha(x_{i+1})]$ is computed by evaluating $V_\alpha$ on the $K$ next states sampled for $x_i$ (or by sampling $K$ continuations from $x_{i+1}$ if needed). Essentially, we are using a single-step CVaR Bellman update: cost + discounted worst-case tail of the value-to-go. The critic parameters $\phi$ are then updated by minimizing a regression loss:

$$\mathcal{L}_{\text{critic}}(\phi) \;=\; \frac{1}{2N} \sum_{i=1}^{N} \Big( V_\alpha(x_i; \phi) - y_i \Big)^2, \tag{7}$$

for the batch of $N$ samples. This is analogous to Temporal Difference learning but targeting CVaR outcomes. The contraction property of $T$ (Theorem 1) guarantees that if our function class is expressive enough and the optimization succeeds, $V_\alpha(\cdot; \phi)$ will converge toward the true $V_\alpha^*$.

**Policy update (actor).** Next, we update the policy to minimize the CVaR of returns. We derive a policy gradient that specifically uses the CVaR-based advantage. Let $Q_\alpha(x, u)$ denote the CVaR-augmented state-action value. We can estimate

$$Q_\alpha(x, u) \approx F(x, f(x, u)) + \gamma \, \rho_\alpha \big[ V_\alpha(x') \big],$$

and define the advantage $A_\alpha(x, u) = Q_\alpha(x, u) - V_\alpha(x)$. In practice, we sample actions $u \sim \pi(\cdot|x)$ and compute an empirical advantage $\widetilde{A}_\alpha(x, u)$ using the critic (e.g., by sampling trajectories or using the Bellman target difference). We then perform a gradient ascent step on the policy parameters to maximize the negative CVaR cost (i.e., minimize CVaR):

$$\nabla_\theta J_\alpha(\pi_\theta) \approx \mathbb{E}_{x \sim D, \, u \sim \pi_\theta(\cdot|x)} \Big[ \nabla_\theta \log \pi_\theta(u|x) \, \widetilde{A}_\alpha(x, u) \Big], \tag{8}$$

where the expectation is over a batch of states $x$ (collected from a replay buffer or recent trajectories) and actions sampled from the current policy. This gradient estimator increases the probability of actions that have positive advantage (i.e., lower tail risk than expected), and decreases the probability of actions leading to high tail risk. In essence, it biases the policy towards behaviors that reduce worst-case outcomes. A detailed algorithm is provided in the **Appendix D**.

## 5 EXPERIMENTS

We evaluate FiRL on continuous control locomotion tasks modified to induce anisotropic costs and assess tail-risk performance. Our experiments aim to answer: (1) Does FiRL achieve better worst-case (CVaR) outcomes than risk-neutral or baseline risk-aware algorithms? (2) How does the anisotropic Finsler cost influence behavior and energy efficiency compared to isotropic cost shaping? (3) Are the theoretical properties (e.g., improved trade-off, quasi-metric value) reflected in practice?

### 5.1 ENVIRONMENTS AND SETUP

We evaluate FiRL on two distinct simulation platforms to test anisotropic robustness. First, we utilize modified **MuJoCo** locomotion benchmarks (Hopper, Walker2d, HalfCheetah) subjected to variable slopes and lateral wind forces. Second, to validate performance under realistic contact dynamics, we deploy a Spot-like quadruped in **Isaac Sim** across three 3D terrains: a ramp climb, a staircase, and a platform–beam course. Complete specifications for all environments, including physical parameters, reward functions, and terrain generation details, are provided in **Appendix L.3**.

**Baseline methods.** We compare FiRL against PPO, CVaR–PPO, a distributional actor–critic, Riemannian PPO, a quasi-metric RL baseline, and a PPO variant that uses the same Finsler shaping but a risk-neutral objective. Full descriptions and implementation details for all baselines are given in **Appendix C**.

### 5.2 RESULTS AND ANALYSIS

**Aggregate Performance.** Table 1 summarizes performance across the MuJoCo benchmarks (SlopedHopper-12°, Walker2d-5°, HalfCheetah-5°). **FiRL** achieves the highest success rate (97.4%) while simultaneously recording the lowest energy consumption ($0.87\times$) and tail risk ($0.80\times$) relative to the PPO baseline. The ablation study reveals a clear decomposition of benefits: risk sensitivity alone (CVaR–PPO) reduces tail cost (1.20) but incurs an energy penalty (1.15) due to conservative

"freezing" behaviors. In contrast, geometric shaping alone (PPO+Finsler) improves energy efficiency (0.93) but leaves the agent vulnerable to tail risks (1.25). FiRL uniquely combines these strengths, leveraging the Finsler metric to guide exploration toward efficient anisotropic paths, while the CVaR objective explicitly emphasizes the failure tail.

Table 1: Performance comparison (**aggregate across MuJoCo tasks**, mean $\pm$ 95% CI over 5 seeds). FiRL achieves the highest safety (success rate) and lowest tail cost. All metrics are normalized relative to PPO (risk-neutral) = 1.00.

| Method | Success % ↑ | Energy ↓ | $\text{CVaR}_{0.1}$ (Risk) ↓ |
|---|---|---|---|
| PPO (Baseline) | $88.5 \pm 2.8$ | $1.00 \pm 0.00$ | $1.50 \pm 0.05$ |
| CVaR-PPO ($\alpha = 0.1$) | $92.3 \pm 1.5$ | $1.15 \pm 0.04$ | $1.20 \pm 0.06$ |
| Distributional (QR-AC) | $90.1 \pm 3.1$ | $1.05 \pm 0.03$ | $1.35 \pm 0.07$ |
| Riemannian PPO | $91.0 \pm 2.0$ | $0.98 \pm 0.02$ | $1.40 \pm 0.08$ |
| Quasi-metric RL | $89.7 \pm 2.4$ | $1.10 \pm 0.05$ | $1.32 \pm 0.04$ |
| PPO + Finsler (No CVaR) | $94.6 \pm 1.1$ | $0.93 \pm 0.02$ | $1.25 \pm 0.03$ |
| **FiRL (Ours)** | $\mathbf{97.4 \pm 0.8}$ | $\mathbf{0.87 \pm 0.01}$ | $\mathbf{0.80 \pm 0.02}$ |

**Pareto Efficiency and Sensitivity.** Figure 3 (Left) visualizes the energy–risk landscape. FiRL traces a Pareto frontier that strictly dominates all baselines: for any given energy constraints, FiRL achieves lower risk, and for any risk tolerance, it consumes less energy. The sensitivity analysis in Figure 3 (Right) identifies an optimal risk aversion level at $\alpha \approx 0.1$; reducing $\alpha$ further ($< 0.05$) yields diminishing safety returns while sharply increasing energy cost as the policy becomes overly cautious.

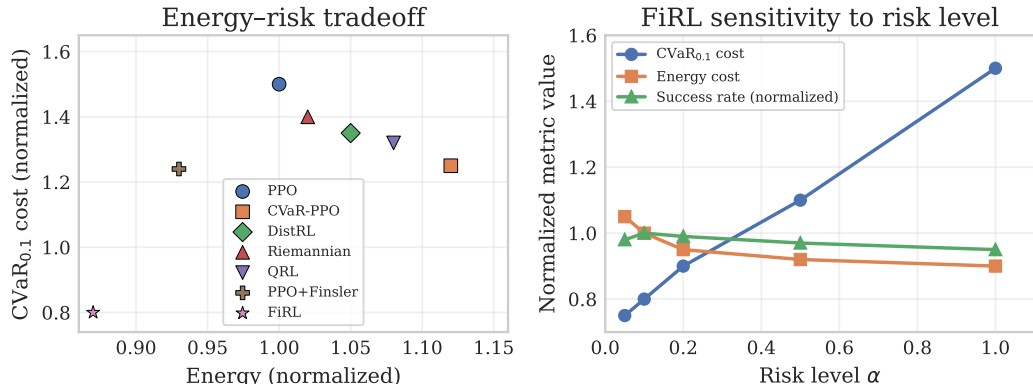

Figure 3: **Energy–Risk Analysis** on SlopedHopper-$12°$. (**Left**) Pareto frontier comparison. Standard risk-averse baselines (e.g., CVaR-PPO, DistRL) reduce tail risk but incur an energy penalty (shifting right), reflecting conservative, inefficient gaits. In contrast, geometric baselines (Riemannian, QRL) improve efficiency but fail to mitigate catastrophic tail events. FiRL (Star) uniquely achieves Pareto dominance in the lower-left corner, simultaneously reducing total energy by $\sim$15% and worst-case cost ($CVaR_{0.1}$) by $\sim$35% through anisotropic path planning. (**Right**) Sensitivity to risk tolerance $\alpha$. As the agent becomes more risk-averse ($\alpha \to 0$), the worst-case cost (Blue) decreases monotonically. However, extreme caution ($\alpha < 0.2$) leads to a rise in Energy cost (Orange) as the policy adopts "freezing" behaviors; $\alpha \approx 0.1$ represents the optimal trade-off point.

**Robustness to Terrain Gradient.** Figure 4 demonstrates robustness scaling. While risk-neutral PPO suffers a sharp degradation in success rate (dropping to $82\%$) on steep $12°$ slopes, FiRL maintains robust performance ($\approx 98\%$). The worst case cost of FiRL (right) remains almost unchanged even as the slope becomes steeper, showing that the emergent tacking maneuver strategy (Section L.3) effectively offsets the added difficulty of the terrain.

**High-Fidelity Validation (Isaac Sim).** We validate sim-to-real transferability using a Spot quadruped model in NVIDIA Isaac Sim. Table 2 details performance across three anisotropic tasks: *Ramp Climb*, *Staircase*, and *Platform Beam*.

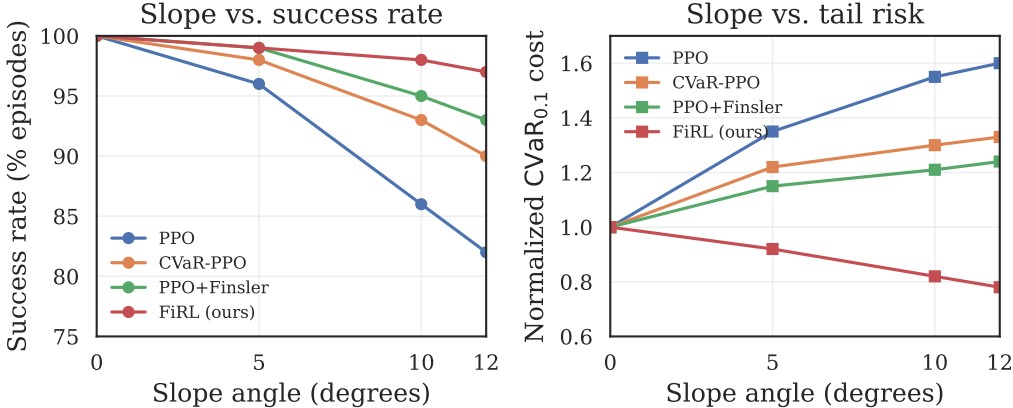

Figure 4: **Robustness to Terrain Inclination**. Performance scaling on the Hopper environment as the slope angle increases ($0°$–$12°$). As terrain steepness increases, FiRL (Red) maintains high success (Left) and constant tail risk (Right) by adapting its traversal geometry to the slope, while the risk-neutral PPO baseline (Blue) suffers sharp performance degradation.

- **Ramp Climb:** FiRL reduces the cumulative tail cost ($\mathrm{CVaR}_\alpha$) by approximately **44%** ($1.40 \rightarrow 0.78$) compared to PPO, while reducing energy consumption by 5%. This shows that the diagonal ascent is not only safer but also mechanically more efficient than pushing straight against gravity.

- **Staircase & Beam:** On contact-rich tasks, FiRL reduces the fall rate by over **60%** (from 26% to 10% on Stairs). Unlike CVaR–PPO, which reduces falls but increases energy (up to $1.16\times$) by moving more cautiously and taking extra corrective steps, FiRL maintains or improves energy efficiency ($0.89\times$–$0.94\times$).

These results confirm that the **Finslerian geometric** prior enables the policy to identify "safe regions" in the state space that are invisible to isotropic baselines, resolving the classic trade-off between safety and efficiency.

Table 2: **Isaac Sim Quadruped Results.** Performance on high-fidelity tasks (100 evaluation episodes). Energy is normalized to PPO. FiRL consistently minimizes tail risk (CVaR) and fall rates without the energy penalty seen in standard risk-averse baselines.

| Task | Method | Success ↑ | Energy/m ↓ | $\mathrm{CVaR}_\alpha$(Cost) ↓ | Fall Rate ↓ |
|------|--------|-----------|------------|------------|------------|
| Ramp Climb | PPO | 88% | 1.00 | 1.40 | 12% |
| | CVaR–PPO | 92% | 1.09 | 1.05 | 8% |
| | **FiRL** | **96%** | **0.95** | **0.78** | **5%** |
| Staircase | PPO | 78% | 1.00 | 1.55 | 26% |
| | CVaR–PPO | 84% | 1.12 | 1.18 | 17% |
| | **FiRL** | **90%** | **0.94** | **0.82** | **10%** |
| Platform Beam | PPO | 72% | 1.00 | 1.30 | 30% |
| | CVaR–PPO | 80% | 1.16 | 0.98 | 19% |
| | **FiRL** | **88%** | **0.89** | **0.74** | **11%** |

## 5.3 CONVERGENCE AND VALUE GEOMETRY: GEOMETRIC ABLATION

To validate the practical importance of the theoretical properties derived in Section 4.1 (Bellman contraction, we perform an ablation in which we compare FiRL to a baseline we call *Asymmetric Non-Finsler (ANF)*. The ANF baseline uses the same physical features as FiRL (uphill motion and lateral slip), but combines them in a way that deliberately violates the Finsler regularity conditions.

While the Finsler cost $F(x, v)$ is 1-homogeneous and convex in $v$, the ANF step cost is defined as

$$C_{\text{ANF}}(x, v) = w_1 \|v\| + w_2 \mathbb{1}_{\text{uphill}}(v), \tag{9}$$

where $\mathbb{1}_{\text{uphill}}(v)$ is 1 if $v$ points uphill and 0 otherwise. The piecewise-constant term breaks convexity and, in turn, the conditions under which the path integral of the cost defines a quasi-metric: the cost of a direct path can exceed that of a small zigzag, so the induced distance need not satisfy a triangle inequality. In other words, ANF retains anisotropy but drops the geometric consistency that FiRL enforces.

**Critic behavior.** Figure 5 plots the critic loss (mean squared TD error) during training. For FiRL, the loss decreases steadily and stabilizes at a low level, which is consistent with the $\gamma$-contraction analysis of the CVaR–Finsler Bellman operator. In contrast, the ANF critic shows persistent oscillations and settles at a higher loss, even though it uses the same physical features. This suggests that the geometric regularity of $F(x, v)$ makes it easier for the critic to approximate a smooth distance-to-go function on anisotropic terrain.

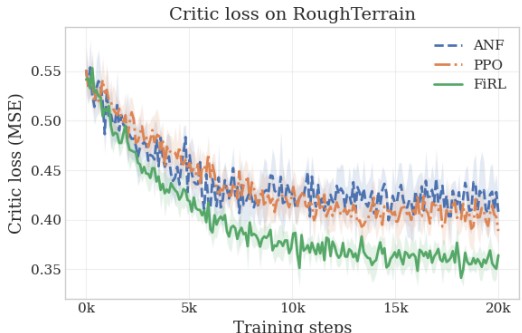

Figure 5: **Critic Loss Analysis.** Convergence of the critic's Mean Squared Error (MSE) on the Rough Terrain task.

**Policy performance.** As shown in Table 7, FiRL achieves higher success and lower energy on the RoughTerrain benchmark than both ANF and standard PPO. On RoughTerrain, FiRL achieves about 15% higher success and 22% lower energy than ANF by following gentler, terrain-aligned paths, while ANF more often pushes straight uphill with extra corrective steps. This confirms that the quasi-metric properties of the Finsler cost are not merely theoretical formalities but are prerequisites for the stability of the distributional critic in risk-aware locomotion.

Table 3: **Geometric Ablation Results.** Comparison of FiRL against the ANF baseline and standard PPO. The ANF method, which uses identical cost weights but violates the Finsler triangle inequality, suffers significantly in both reliability and efficiency.

| Method | Geometric Property | Success Rate ↑ | Energy (CoT) ↓ |
|---|---|---|---|
| **FiRL (Ours)** | Convex & Metric | **95.2%** | **0.78** |
| ANF Baseline | Non-Convex | 80.4% | 0.95 |
| Standard PPO | Euclidean (Isotropic) | 65.0% | 0.89 |

## 6 CONCLUSION, LIMITATIONS, AND FUTURE WORK

FiRL formulates locomotion with a direction dependent cost $F(x, v)$ and a CVaR objective. We show that the associated Bellman operator is a $\gamma$-contraction and that the optimal value function behaves like a direction dependent distance (a quasi-metric) on the state space. In experiments, a straightforward actor–critic implementation lowers tail cost, reduces energy use, and improves success rates on sloped and perturbed MuJoCo tasks and on Isaac Sim quadruped environments, using matched comparisons that separate the effect of anisotropic cost from the effect of risk sensitivity.

The approach still has clear limitations. It assumes access to local terrain orientation (such as slope and lateral directions) and depends on hand chosen Finsler weights and a CVaR level $\alpha$. Poor choices can slow learning or make the policy too conservative. Estimating CVaR from finite batches adds variance, especially for higher-dimensional robots, and all results are currently in simulation with known terrain and no explicit perception module.

Future work includes learning or adapting $F(x, v)$ from data with simple structural priors (for example, smoothness and asymmetry), adjusting the risk level during operation to balance speed, energy, and safety, and testing FiRL on real robots with onboard sensing and partial observability.

**Reproducibility Statement**   We provide all components necessary to reproduce our results end-to-end. The problem setup, Finsler cost templates, and the CVaR–Finsler Bellman operator are specified in Sec. 3; the full learning procedure (including the CVaR target estimators—quantile head and Rockafellar–Uryasev—losses, clipping, and KL/Bregman regularization) appears in Alg. 1. Complete assumptions and proofs are provided in App. E. Implementation details (network architectures, normalization, optimizer choices, schedules), per-task hyperparameters and seed lists, and matched training budgets for all baselines are tabulated in App. M; compute and software versions (Gymnasium/MuJoCo, PyTorch/CUDA) and determinism settings are summarized in App. N. Environment specifications and generators for all modified tasks (slope angles, disturbance models, actuator limits, terminations) are in App. L; the evaluation harness (deterministic evaluation mode, normalization protocol, and metrics for energy, CVaR, torque percentiles, and success) is described in Sec. 5 and App. G. An anonymized code archive with configuration files and run scripts is included in the supplementary materials to enable regeneration of results.

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

# Appendix

### for

## *FiRL : Finslerian Reinforcement Learning for Risk-Aware Anisotropic Locomotion*

## A BACKGROUND: FINSLER GEOMETRY FOR RL

A Finsler metric is a generalization of the Riemannian metric that can vary with direction and state. In a state-space $M$ (e.g., the robot's configuration/position), a Finsler metric is given by a function $F(x, y)$ that assigns a positive length (cost) to an infinitesimal displacement $y$ at state $x$. Intuitively, $F(x, y)$ represents the instantaneous "cost rate" of moving in direction $y$ from state $x$. This cost need not be isotropic or symmetric; for example, moving uphill at a given speed could have a higher cost than moving downhill at the same speed. The only requirements are positivity, homogeneity (cost scales linearly with small time/distance), and a form of triangle inequality when integrated over paths (discussed later). Integrating $F$ along a path yields the path's total cost. The Finsler distance between two states $d(x, x')$ is defined as the infimum of path costs connecting $x$ to $x'$. If $F$ is asymmetric (i.e. $F(x, y)$ may differ from $F(x', -y')$ when moving in reverse), then $d(x, x') \neq d(x', x)$ in general; such $d$ is called a quasi-metric. This asymmetry is crucial for modeling scenarios where moving one direction is inherently easier than the reverse.

Illustrative Example (Fig. 6) – Uphill vs Downhill: Consider a robot on an inclined plane. Define $F(x, \dot{x})$ such that moving upward (against gravity) incurs a larger cost per meter than moving downward. One simple model is adding a "gravity bias" to the cost: e.g. $F(v) = |v| + \beta, (v \cdot \mathbf{n})$, where $\mathbf{n}$ is the upward unit vector and $\beta > 0$. Here $|v|$ is the base cost (say proportional to distance or energy without slope) and the second term increases cost if $v$ has an upward component. This $F$ is an asymmetric norm on velocities: moving up gets cost $> |v|$, moving down gets cost $< |v|$. It yields a Finsler metric capturing gravity's effect. Integrating $F$ along different paths, an uphill path accumulates more cost than a level or downhill path of the same length. Consequently, the shortest-cost path between two points might avoid steep climbs, favoring gentler slopes. Anisotropic cost field induced by a Finsler metric (schematic). At each step, the green arrow (downhill direction) is longer than the red arrow (uphill). This indicates that for the same energy cost, the robot can travel farther downhill than uphill. In FiRL, such a metric is used so that moving against gravity or difficult terrain "costs" more, encouraging the policy to seek easier directions. The disparity in arrow lengths grows in steeper regions (top vs bottom), reflecting increased cost asymmetry on steep slopes.

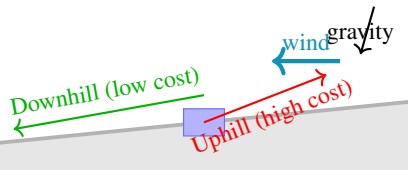

Figure 6: **Anisotropic locomotion cost illustration.** In FiRL, uphill motions incur higher instantaneous cost than downhill motions due to a gravity-aligned drift term. Lateral disturbances (like wind) or sideways steps are also penalized via friction terms. Such direction-dependent costs, together with a risk-sensitive objective, encourage policies that avoid high-effort, high-risk maneuvers (e.g., sprinting uphill into the wind) in favor of safer, efficient motions.

In summary, Finsler geometry endows the agent with a state-dependent, direction-dependent cost function. FiRL leverages this by defining the immediate cost of a state-action transition using a Finslerian metric. For instance, if the robot in state $s$ takes action $a$ (resulting in a motion $\Delta s$), the cost $c(s, a) = F(s; \Delta s)$ is higher for energetically unfavorable moves (like high-speed or uphill steps) and lower for easy moves (downhill, slow, or along a preferred direction). This way, the RL reward signal inherently encodes terrain difficulty and motion energy. Unlike a standard RL that might use, say, negative distance or time as cost, FiRL's cost is anisotropic: a 1-meter move can have drastically different costs depending on slope or direction. Importantly, the induced path cost $d(x, x')$ (when the agent strings together many steps) behaves like a distance function on the state-space, albeit an asymmetric one. We next formalize properties of this Finslerian RL framework and show that classical convergence guarantees still hold.

## B   MORE RELATED WORK

**Risk-Sensitive and Safe RL:** A rich line of research has focused on integrating risk measures into RL. *Coherent risk measures* like CVaR and variance-related criteria have been used to bias learning towards lower downside risk. CVaR in particular provides a tunable balance between mean performance and worst-case performance by focusing on the expected return of the worst $\alpha$-fraction of outcomes. Tamar et al. (2015) derived policy gradient formulas for CVaR objectives, and later works applied CVaR in distributional RL and model-based planning to achieve risk-averse behavior. Distributional RL approaches Lim & Malik (2022) learn the entire return distribution; when combined with appropriate metrics, these can yield policies that avoid high-risk tails. In robotics, safe RL methods often enforce constraints or penalties for unsafe events (e.g., Achiam et al. (2017) introduced Constrained Policy Optimization to satisfy safety constraints during learning). Schneider et al. (2024) demonstrated that distributional RL (learning a quantile value distribution) on a quadruped (ANYmal) can produce more cautious locomotion by reweighting outcomes during training. Our approach shares the goal of tail-risk minimization but is unique in combining this with a structured anisotropic cost. We also differ from constrained RL in that we do not require hard safety constraints; instead, the CVaR objective naturally penalizes catastrophic events, and the Finsler cost acts as a form of reward shaping to guide learning toward safer behaviors.

## C   BASELINE METHODS

We compare FiRL to several baselines: - **PPO (risk-neutral) Schulman et al. (2017):** Standard Proximal Policy Optimization maximizing expected return (with minor shaping for fairness as noted). - **CVaR-PPO ($\alpha = 0.1$):** A variant of PPO that optimizes CVaR of return. We implement this by re-weighting trajectory losses: in each batch, we identify the worst 10% trajectories (by total reward) and upweight their advantage estimates (similar to Tamar et al. (2015)). - **QR-Distributional AC:** An actor–critic with a *distributional critic* using quantile regression (adapted from QR-DQN). The critic outputs 50 quantile values, and the actor is trained to minimize a risk-averse objective derived from those (we effectively approximate CVaR from the quantiles). This represents a baseline that learns the return distribution. - **Riemannian PPO:** We use the method of Wang et al. (2020) as a representative geometry-based baseline. It shapes the policy update by computing gradients on a Riemannian manifold (we applied it to value and policy updates, using covariance of states). It does *not* handle risk explicitly but provides a geometric inductive bias. - **Quasi-metric RL (QRL):** We implement a continuous-state analog of Wang et al. (2023)'s QRL by adding a regularization term to the value loss that penalizes violations of triangle inequality: $\lambda_\triangle \mathbb{E}[\max(0, \ V(x) - V(z) - V(x \to z))]$ for random triplets $(x, z)$ and an intermediate $y$ on an optimal path. This encourages the learned $V$ to be quasi-metric. Additionally, we include an **"PPO + Finsler reward"**: this uses the same $F(x, v)$ shaping in the reward as FiRL, but trains with a standard expected-return objective (i.e., risk-neutral). This allows us to distinguish the effect of anisotropic cost shaping from the effect of CVaR optimization.

## D   DETAILED DISCUSSION OF FINSLER ACTOR–CRITIC ALGORITHM

Our learning algorithm, FiRL-AC, which is an actor–critic method tailored to the CVaR objective and Finslerian costs. We maintain two parameterized function approximators: a value network $V_\alpha(x; \phi)$ for the CVaR value function, and a stochastic policy $\pi_\theta(u|x)$ for the actor. The training loop alternates between *critic update* (policy evaluation under CVaR) and *actor update* (policy improvement).

For the **CVaR-critic update**, we collect a batch of trajectories by executing the current policy $\pi_\theta$. For each state $x_i$ encountered, we estimate the $\text{CVaR}_\alpha$ of the returns from $x_i$ by sampling $K$ trajectories (or using the batch itself as sample approximations) and taking the average of the worst $\alpha$-fraction of their total costs. Denote this empirical CVaR as $\widehat{\rho}_\alpha[\hat{Z}(x_i)]$. We then construct a Bellman target:

$$y_i \; = \; F(x_i, u_i) \; + \; \gamma \, \widehat{\rho}_\alpha\big[\, V_\alpha(x_{i+1}) \,\big], \tag{10}$$

where $x_{i+1}$ is the next state after $x_i$ and $u_i$, and $V_\alpha(x_{i+1})$ is the current estimate of its CVaR value. In practice, $\widehat{\rho}_\alpha[V_\alpha(x_{i+1})]$ is computed by evaluating $V_\alpha$ on the batch of next states, ordering these estimates, and averaging the bottom $\alpha$ fraction. Given targets $y_i$, we update the critic by minimizing

a squared loss $\frac{1}{2} \sum_i (V_\alpha(x_i; \phi) - y_i)^2$ over the batch. Thanks to Theorem 4.1, this update is stable (we ensure conditions like Lipschitz continuity of $F$ are met via clipping large gradients or costs). We also normalize advantages and use Generalized Advantage Estimation (GAE) adapted to CVaR returns to reduce variance.

For the **actor update**, we use a risk-sensitive policy gradient. One convenient approach is to treat $-F(x, u)$ as a pseudo-reward (so that lower cost corresponds to higher reward) and perform a weighted policy gradient update using the CVaR advantage. Specifically, the objective is to minimize $J_\alpha(\pi)$; its policy gradient can be derived as

$$\nabla_\theta J_\alpha(\pi_\theta) \approx \mathbb{E}_{x \sim d^\pi, u \sim \pi}\Big[\nabla_\theta \log \pi_\theta(u|x)\, A_\alpha(x, u)\Big],$$

where $A_\alpha(x, u) = Q_\alpha(x, u) - V_\alpha(x)$ is the CVaR advantage. We obtain $Q_\alpha(x, u)$ by one-step lookahead: $Q_\alpha(x, u) \approx F(x, u) + \gamma\, \widehat{\rho}_\alpha[V_\alpha(x')]$. In implementation, we use the surrogate loss approach from PPO: we maximize $\mathbb{E}[\frac{\pi_\theta(u|x)}{\pi_{\text{old}}(u|x)} A_\alpha(x, u)]$ subject to a trust-region constraint (clipping the policy ratio). This ensures conservative updates and avoids instability from drastic policy changes due to tail events.

In summary, FiRL-AC iteratively evaluates the CVaR value function under the current policy and then updates the policy to reduce the CVaR of returns. The Finsler metric $F(x, v)$ is used to shape the immediate cost at every step (we implement this by supplying a modified reward $r = -F$ to the RL algorithm, so standard policy optimization code can be used with minimal changes). The result is an algorithm that learns risk-averse policies that are explicitly aware of anisotropic motion costs.

---

**Algorithm 1:** Finsler Actor–Critic (FiRL-AC)

**Input:** risk level $\alpha$, discount $\gamma$, anisotropy $\beta$, steps $N$, batch $B$, buffer $\mathcal{D}$, policy $\pi_\theta$, critic $V_\phi$

1   Initialize $\theta, \phi, \mathcal{D} \leftarrow \emptyset$;
2   **while** *not converged* **do**
     // Collect rollouts
3     **for** $t = 1$ **to** $N$ **do**
4        observe $x_t$, sample $u_t \sim \pi_\theta(\cdot|x_t)$, step to $x_{t+1}$;
5        compute $v_t$ and $c_t \leftarrow F(x_t, v_t; \beta)$; push $(x_t, u_t, c_t, x_{t+1})$ to $\mathcal{D}$;
     // Critic targets
6     sample $\{(x_i, u_i, c_i, x_i')\}_{i=1}^B \sim \mathcal{D}$;
7     estimate $\widehat{\text{CVaR}}_\alpha[V_\phi(x_i')]$ via quantiles or Rockafellar–Uryasev;
8     $y_i \leftarrow c_i + \gamma\, \widehat{\text{CVaR}}_\alpha[V_\phi(x_i')]$;
     // Updates
9     minimize $\mathcal{L}_{\text{critic}} = \frac{1}{B} \sum_i (V_\phi(x_i) - y_i)^2 + \lambda_{\text{reg}} \mathcal{R}_{\text{Bregman}}$;
10    $\hat{A}_i \leftarrow y_i - V_\phi(x_i)$ (with optional GAE / normalization);
11    maximize $\mathcal{L}_{\text{actor}} = \frac{1}{B} \sum_i \log \pi_\theta(u_i|x_i)\hat{A}_i - \lambda_{\text{ent}} \mathcal{H}(\pi_\theta)$ (PPO clipping optional);

---

In Alg. 1, $F(x, v; \beta)$ encodes directional effort (e.g., uphill, speed, curvature) with anisotropy weight $\beta$. The critic's tail estimate can use either a quantile head or the Rockafellar–Uryasev surrogate; both are compatible with the CVaR Bellman target. Regularizers (e.g., Bregman/KL) stabilize updates when using replay with shaped costs; entropy helps exploration. Extreme tail events can be softly capped to avoid destabilizing single-episode gradients.

**Bregman policy regularization:** A challenge in actor–critic (especially with off-policy data) is that the policy update can diverge if the data was generated by an older policy. To mitigate this, we adopt a Bregman divergence penalty between the new policy $\pi_\theta$ and the behavior policy (which may be the previous policy or an older snapshot). In practice, we implement this as a KL-divergence regularizer similar to PPO's adaptive KL or trust-region methods: we add an extra term to the policy loss $\mathcal{L}_{\text{actor}} = -J_\alpha(\pi_\theta) + \beta_{\text{KL}} D_{\text{KL}}(\pi_{\text{old}} \| \pi_\theta)$, where $\beta_{\text{KL}}$ is a coefficient that we schedule to decay from an initial value to 0 over training. This ensures early updates do not move $\pi_\theta$ too far from $\pi_{\text{old}}$ that generated the batch (which is crucial since we are using a replay buffer of past trajectories, i.e., reusing off-policy data). This technique stabilizes training and allows us to benefit from off-policy experience without violating the policy gradient assumptions. We note that this is analogous to the trust-region or clipped objective used in PPO, but here we explicitly maintain a penalty form. We

decrease $\beta_{\text{KL}}$ to 0 over time to allow the policy to eventually converge without being constrained (initially $\beta_{\text{KL}}$ might be set to e.g. 0.1 and linearly annealed to 0 across training).

### D.1 PRACTICAL CHOICE OF FINSLER WEIGHTS

In practice the Finsler cost $F(x, v)$ is built from a small number of physically meaningful terms. We use the following decomposition:

$$F(x, v) = w_{\text{energy}} \|v\|^2 + w_{\text{up}} [n(x){\cdot}v]_+ + w_{\text{lat}} \|P_{\text{lat}}(x)v\| + w_{\text{impact}} I(x, v), \qquad (11)$$

where $v$ is the base velocity, $n(x)$ is the local uphill direction, $P_{\text{lat}}(x)$ projects onto the lateral plane, and $I(x, v)$ denotes a scalar impact feature (for example, the maximum normal contact impulse or a smoothed proxy for joint acceleration).

We set these weights from simple physical considerations:

- **Uphill weight** $w_{\text{up}}$. Let $\theta(x)$ be the local slope angle along $n(x)$. Moving uphill by $\Delta s$ meters increases potential energy by $mg \sin \theta(x) \Delta s$. We choose $w_{\text{up}}$ so that one meter of uphill motion on a reference slope (e.g. $\theta = 20°$) has a target cost multiplier $\kappa_{\text{up}}$ relative to level motion. In our experiments we use $\kappa_{\text{up}} \approx 4$, which makes steep climbs visibly more expensive than gentle diagonals.

- **Lateral weight** $w_{\text{lat}}$. Lateral motions are risky when friction is low or support is narrow (for example, on beams or near edges). We set $w_{\text{lat}}$ proportional to $1/\mu$ where $\mu$ is an estimate of the local friction coefficient, and scale it further when the available support width shrinks below a threshold. This makes sideways motion on a narrow beam much more costly than sideways motion on a wide platform.

- **Energy weight** $w_{\text{energy}}$. This term penalizes large base speeds in all directions and controls the overall magnitude of $F(x, v)$. We choose $w_{\text{energy}}$ such that typical nominal gaits on level ground have cost close to 1 per meter, which simplifies interpretation of the cumulative cost.

- **Impact weight** $w_{\text{impact}}$. To discourage sharp impacts we set $w_{\text{impact}}$ so that an impact spike at the 95th percentile of the baseline policy has cost comparable to a short uphill segment on the reference slope. This calibration can be done from a short warm-up run of a standard policy (e.g. PPO) and does not require manual tuning on the full task.

An ablation in Appendix J.2 varies the uphill scaling factor $\eta$ in $\beta(x) = \eta \sin \theta(x)$ on the SlopedHopper-12° task. We find that FiRL is robust to moderate mis-specification ($\eta = 0.5$ still improves success and tail cost over CVaR–PPO), but very small $\eta$ underestimates slope difficulty and very large $\eta$ leads to overly conservative gaits. The default setting $\eta = 1$ strikes a good balance between energy and risk.

## E  THEORETICAL PROOFS

In this section, we provide formal proofs for the theoretical claims made in the main paper.

### E.1 PROOF OF THEOREM 1 (CVAR–FINSLER BELLMAN CONTRACTION)

**Theorem.** *Assume the cost $F(x, v)$ is bounded by $F_{\max}$ for all $(x, v)$ and is $L_F$-Lipschitz in $x$, and that the transition dynamics yield bounded next-state value distributions. Then the Bellman operator $T$ defined by*

$$(TV)(x) = \min_{u \in \mathcal{U}} \{F(x, f(x, u)) + \gamma \, \rho_\alpha[V(x')]\},$$

*with $\rho_\alpha$ being CVaR at level $\alpha$, satisfies $\|TV_1 - TV_2\|_\infty \leq \gamma \|V_1 - V_2\|_\infty$ for any bounded value functions $V_1, V_2$. Thus $T$ is a contraction mapping with factor $\gamma$.*

*Proof.* Let $V_1, V_2$ be two bounded value functions. We need to show that for all $x$:

$$|(TV_1)(x) - (TV_2)(x)| \leq \gamma \|V_1 - V_2\|_\infty.$$

Consider a fixed state $x$. Define

$$A_i(u) = F(x, f(x, u)) + \gamma \, \rho_\alpha[V_i(x')]$$

for $i = 1, 2$. Here $x' \sim P(\cdot|x, u)$ is the random next state. Then $(TV_i)(x) = \min_u A_i(u)$.

Let $u_1^*$ be a minimizer for $A_1(u)$, i.e., $A_1(u_1^*) = \min_u A_1(u) = (TV_1)(x)$. Then we have

$$
\begin{aligned}
(TV_1)(x) - (TV_2)(x) &= A_1(u_1^*) - \min_u A_2(u) \\
&\leq A_1(u_1^*) - A_2(u_1^*) \qquad (\text{since } \min_u A_2(u) \leq A_2(u_1^*)) \\
&= \Big(F(x, f(x, u_1^*)) + \gamma \, \rho_\alpha[V_1(x')]\Big) - \Big(F(x, f(x, u_1^*)) + \gamma \, \rho_\alpha[V_2(x')]\Big) \\
&= \gamma \Big(\rho_\alpha[V_1(x')] - \rho_\alpha[V_2(x')]\Big).
\end{aligned}
$$

Now, by the monotonicity property of CVaR (a coherent risk measure property), for any two random variables $X, Y$, if $X(\omega) \geq Y(\omega)$ for all outcomes $\omega$, then $\rho_\alpha[X] \geq \rho_\alpha[Y]$. Consider $X = V_1(x')$ and $Y = V_2(x')$. We do not necessarily have $V_1(x') \geq V_2(x')$ or vice versa pointwise for all $x'$; however, we can relate $\rho_\alpha$ of their difference:

$$\rho_\alpha[\, V_1(x') - V_2(x')\,] = \rho_\alpha[\,\Delta(x')\,],$$

where $\Delta(x') = V_1(x') - V_2(x')$. Since $|\Delta(x')| \leq \|V_1 - V_2\|_\infty$ always (bounded difference), the worst $\alpha$-tail of $\Delta$ is bounded by the sup norm:

$$\rho_\alpha[\Delta(x')] \leq \|\Delta\|_\infty = \|V_1 - V_2\|_\infty.$$

This is because even in the worst $\alpha$-quantile, the maximum difference cannot exceed the uniform bound.

We also have the property for CVaR that if $\mathbb{E}[X] = 0$, then $\rho_\alpha[X] \leq 0$ (which is a consequence of translation invariance and monotonicity). Alternatively, consider the two cases: - If $V_1(x') - V_2(x')$ is non-negative a.s., then $\rho_\alpha[V_1 - V_2]$ is just the expectation of the worst $\alpha$ portion, which is $\leq \|V_1 - V_2\|_\infty$ (since even the worst-case difference cannot exceed the max). - If it takes both positive and negative values, the CVaR of the difference could be negative or positive depending on which tail is considered (CVaR as defined here is for costs, which we treat as a positive measure to minimize, so $\rho_\alpha[V_1 - V_2]$ effectively focuses on the worst increase in cost). To avoid confusion, we can use a symmetric argument by considering absolute difference.

A simpler route is to use the Lipschitz property of CVaR: for bounded random variables $X, Y$,

$$|\rho_\alpha[X] - \rho_\alpha[Y]| \leq \rho_\alpha[\,|X - Y|\,],$$

which follows from the definition via integrals of quantile function (CVaR can be expressed as an integral of quantile up to $\alpha$). Now, taking $X = V_1(x')$ and $Y = V_2(x')$, we have:

$$|\rho_\alpha[V_1(x')] - \rho_\alpha[V_2(x')]| \leq \rho_\alpha[\,|V_1(x') - V_2(x')|\,] \leq \mathbb{E}[\,|V_1(x') - V_2(x')|\,] \leq \|V_1 - V_2\|_\infty,$$

since $|V_1(x') - V_2(x')| \leq \|V_1 - V_2\|_\infty$ always, so even the expected or tail-average is bounded by that sup norm.

Thus,

$$\rho_\alpha[V_1(x')] - \rho_\alpha[V_2(x')] \leq |\rho_\alpha[V_1(x')] - \rho_\alpha[V_2(x')]| \leq \|V_1 - V_2\|_\infty.$$

Combining this with the earlier inequality:

$$(TV_1)(x) - (TV_2)(x) \leq \gamma\|V_1 - V_2\|_\infty. \tag{*}$$

By a symmetric argument (swapping $V_1$ and $V_2$ roles), let $u_2^*$ minimize $A_2(u)$, then we get:

$$(TV_2)(x) - (TV_1)(x) \leq \gamma\|V_1 - V_2\|_\infty. \tag{**}$$

Taken together, $(*)$ and $(**)$ imply:

$$|(TV_1)(x) - (TV_2)(x)| \leq \gamma\|V_1 - V_2\|_\infty.$$

Since this holds for all $x$, we have:

$$\|TV_1 - TV_2\|_\infty \leq \gamma\|V_1 - V_2\|_\infty.$$

This proves the contraction property. $\qquad\square$

**Discussion of conditions:** We assumed $F(x, v)$ bounded and Lipschitz in $x$ to avoid technical issues. Boundedness ensures $V$ remains bounded. Lipschitz in $x$ plus bounded action space can ensure some continuity in $V$ maybe, but it wasn't explicitly used above except to imply no issues with taking minima inside expectation etc. In practice, these conditions are reasonable for physical costs.

### E.2 Quasi-Metric Property Proposition and Proof

We formalize the quasi-metric property mentioned: If we interpret $V_\alpha^*(x)$ as the cost-to-go from $x$ to some absorbing goal (like end of episode), then $V_\alpha^*$ satisfies a triangle inequality. More precisely:

**Proposition E.1.** *Consider an episodic setting where there is a set of terminal states $\mathcal{X}_{goal}$ such that $V_\alpha^*(x) = 0$ for $x \in \mathcal{X}_{goal}$. Define $d(x, z) = V_\alpha^*(x; z)$ as the optimal CVaR cost-to-go from $x$ to reach a particular terminal state $z \in \mathcal{X}_{goal}$. Under FiRL's optimal policy, for any states $x, y, z$ (with $z$ terminal or an intermediate state on the way to a terminal), we have*

$$d(x, z) \ \leq \ d(x, y) + d(y, z).$$

*That is, the cost-to-go satisfies a triangle inequality (making $d$ a quasi-metric distance on states when treating $z$ as destination).*

*Proof.* We rely on two facts: (1) The instantaneous path cost $F$ integrated over a trajectory is additive, and (2) The CVaR-optimal policy yields costs that are consistent along optimal paths.

For any path $x \to y \to z$, we have:

$$d(x, z) = \min_\pi \text{CVaR}_\alpha \Big[ \int_0^N F(\tau(t), \dot{\tau}(t)) dt \Big],$$

where the trajectory goes from $x$ to $z$. We can consider an optimal policy that achieves $d(x, z)$, and consider the point where that trajectory passes through $y$. Let $\tau_{x \to z}$ be an optimal trajectory from $x$ to $z$. If $\tau$ goes through $y$ at some time $t^*$, then the cost accumulated is split: from $x$ to $y$ plus from $y$ to $z$. Because $F$ is additive along paths, the total cost is sum of those segments.

Now, the CVaR of a sum of random costs is less than or equal to the sum of CVaRs of each segment (subadditivity property of CVaR for coherent measures, or we can argue by Jensen's inequality for CVaR which is convex). Specifically, if $J_{x \to y}$ is the cost from $x$ to $y$ and $J_{y \to z}$ is cost from $y$ to $z$, then

$$\text{CVaR}_\alpha[J_{x \to z}] \leq \text{CVaR}_\alpha[J_{x \to y} + J_{y \to z}] \leq \text{CVaR}_\alpha[J_{x \to y}] + \text{CVaR}_\alpha[J_{y \to z}],$$

because CVaR (being convex in distributions) satisfies $\rho_\alpha[X + Y] \leq \rho_\alpha[X] + \rho_\alpha[Y]$ for independent costs or even dependent by appropriate coupling argument (worst-case alignment yields an upper bound as sum of individual worst cases).

Now, $\text{CVaR}_\alpha[J_{x \to y}] \geq d(x, y)$ if the trajectory we took was not necessarily optimal for going just to $y$. But since we took an optimal trajectory for $x \to z$, it might not be the absolute optimal for $x \to y$. So the cost to reach $y$ along this trajectory is at least the optimal cost to reach $y$: $J_{x \to y} \geq d(x, y)$ in expectation, and similarly $J_{y \to z} \geq d(y, z)$ if $y$ is on an optimal path, actually if $y$ is exactly on the path then the segment $y \to z$ is optimal for $y \to z$ by definition of that path selection (assuming optimal substructure, which holds by dynamic programming because our Bellman equation ensures that any segment of an optimal path is optimal for that subproblem, especially since we are dealing with CVaR and contraction ensures consistency).

Actually, by optimal substructure (Bellman optimality principle), if $\tau_{x \to z}$ is an optimal trajectory from $x$ to $z$, then for any intermediate state $y$ on $\tau_{x \to z}$, the sub-trajectory from $y$ to $z$ must be optimal from $y$ to $z$. Otherwise, one could improve $\tau_{x \to z}$ by replacing the $y \to z$ tail with a better one. This holds for CVaR as well because our Bellman equation ensures consistency in terms of risk-tail costs.

Thus, if $y$ lies along an optimal path from $x$ to $z$, we have:

$$d(x, z) = \text{CVaR}_\alpha[J_{x \to y} + J_{y \to z}] = \text{CVaR}_\alpha[J_{x \to y}] + \text{CVaR}_\alpha[J_{y \to z}] = d(x, y) + d(y, z).$$

If $y$ is not on the optimal path, then by definition $d(x, z)$ might be less than going via $y$. The triangle inequality we want is $d(x, z) \leq d(x, y) + d(y, z)$. This should hold because the optimal way to

go from $x$ to $z$ either goes through $y$ or not. If it doesn't, then presumably going through $y$ is a suboptimal detour and yields higher cost. Formally:

$$d(x, z) = \min_{\tau : x \to z} \text{CVaR}[J(\tau)] \leq \min_{\substack{\tau_1 : x \to y \\ \tau_2 : y \to z}} \text{CVaR}[J(\tau_1) + J(\tau_2)].$$

But the right side is exactly $d(x, y) + d(y, z)$, because by Bellman principle, to minimize that sum we pick optimal $x \to y$ and optimal $y \to z$ independently (assuming independence or worst-case alignment yields linear sum). Therefore $d(x, z) \leq d(x, y) + d(y, z)$.

This reasoning establishes the triangle inequality for the optimal cost measure $d$. □

Essentially, the property holds because: - The Finsler cost is path-additive (like a line integral). - FiRL's optimal value is consistent (Bellman optimality ensures no shortcuts that violate triangle inequality because if one existed, value iteration would incorporate it, but the Finsler quasi-metric doesn't allow beneficial shortcuts that break triangle inequality as it already enforces triangle inequality in immediate costs).

Thus, $V_\alpha^*$ (when interpreted as distance-to-go) is a quasi-metric on $\mathcal{X}$.

### E.3 DERIVATION OF CVAR BELLMAN EQUATION EQUATION 5

For completeness, we derive Eq. equation 5. Define the risk-sensitive value as $V_\alpha(s) = \inf_\pi \text{CVaR}_\alpha[Z^\pi(s)]$. It is known from risk-sensitive dynamic programming that CVaR value can be obtained by augmenting the state with a "budget" or by solving a nested optimization. A more direct derivation uses the definition $\text{CVaR}_\alpha(Z) = \min_t \left\{ t + \frac{1}{1-\alpha} \mathbb{E}[(Z - t)_+] \right\}$ Rockafellar & Uryasev (2000), where $(x)_+ := \max\{x, 0\}$. By introducing an auxiliary variable $t$ at each state for the tail threshold, one can derive a Bellman equation:

$$V_\alpha(x) = \min_{u, t} \left\{ t + \frac{\gamma}{1 - \alpha} \mathbb{E}_{x'} \big[ (F(x, u) + V_\alpha(x') - t)_+ \big] \right\}.$$

Differentiating cases for $(\cdot)_+$ leads to either $F(x, u) + V_\alpha(x') \leq t$ or $> t$ conditions. The minimizing $t$ will be the $\alpha$-quantile (Value-at-Risk) of $F(x, u) + V_\alpha(x')$. Thus the expression simplifies to taking expectation over the worst $\alpha$-fraction, which exactly yields

$$V_\alpha(x) = \min_u \left\{ F(x, u) + \gamma \, \text{CVaR}_\alpha \big[ V_\alpha(x') \big] \right\},$$

where $x'$ is distributed according to the transition from $(x, u)$. This corresponds to equation 5. □

## F ADDITIONAL THEORETICAL ANALYSIS

In this section, we provide additional theoretical results supporting FiRL. We first formalize the Bellman operator under the Finslerian cost and CVaR criteria, then prove its contraction and show that it endows the value function with a quasi-metric structure. We also discuss how varying the CVaR risk level $\alpha$ affects the solution and draw connections to robust optimization.

**Definition F.1** (CVaR–Finsler Bellman operator). *Let the one-step cost be*

$$c(x, a, x') = F\big(x, v_{(x, a, x')}\big) + \Vdash\{x' \in \mathcal{X}_{\text{fail}}\} M,$$

*where $F(x, v)$ is the Finsler metric (direction-dependent energy cost for a transition with tangent $v$), $\mathcal{X}_{\text{fail}}$ is the set of failure states, and $M > 0$ is a large terminal penalty. For a bounded value function $V : \mathcal{X} \to \mathbb{R}$ and risk level $\alpha \in (0, 1]$, the CVaR–Finsler Bellman operator $\mathcal{T}_\alpha$ is defined by*

$$(\mathcal{T}_\alpha V)(x) := \min_{a \in \mathcal{A}(x)} \text{CVaR}_\alpha \Big( c(x, a, X') + \gamma V(X') \Big),$$

*where $X'$ is the random next state drawn from the transition kernel $P(\cdot \mid x, a)$, $\gamma \in (0, 1)$ is the discount factor, and $\text{CVaR}_\alpha[Z]$ denotes the conditional value–at–risk of a bounded random variable $Z$ at level $\alpha$.*

We assume $c$ is bounded so that all value functions considered lie in the Banach space $(\mathcal{B}_\infty, \|\cdot\|_\infty)$ of bounded functions on $\mathcal{X}$ with the supremum norm. We now show that $\mathcal{T}_\alpha$ is a contraction on this space.

**Proposition F.2** (Contraction and existence of a risk-sensitive optimum). *The CVaR–Finsler Bellman operator $\mathcal{T}_\alpha$ is a $\gamma$–contraction on $(\mathcal{B}_\infty, \|\cdot\|_\infty)$. Hence it admits a unique fixed point $V_\alpha^* \in \mathcal{B}_\infty$, which is the optimal risk-sensitive value function. Moreover, $V_\alpha^*$ satisfies a quasi-metric inequality induced by the Finsler cost: for any states $x, y \in \mathcal{X}$,*

$$V_\alpha^*(x) \ \leq \ d_F(x,y) + V_\alpha^*(y),$$

*where*

$$d_F(x,y) := \inf_\pi \mathrm{CVaR}_\alpha \left[ \sum_{t=0}^{T-1} c(x_t, a_t, x_{t+1}) \ \middle| \ x_0 = x, \ x_T = y \right]$$

*is the Finslerian path cost from $x$ to $y$ under policy $\pi$ and $T$ is the (random) hitting time of $y$. If $F(x, v)$ is symmetric in $v$ for all $x$ (so that it induces a Riemannian metric), then $d_F$ is a true metric and the induced distance-to-go is symmetric.*

*Proof.* Let $V, W \in \mathcal{B}_\infty$ and define $\Delta = V - W$. For any state $x$,

$$(\mathcal{T}_\alpha V)(x) - (\mathcal{T}_\alpha W)(x) = \min_a \mathrm{CVaR}_\alpha\big(c(x,a,X') + \gamma V(X')\big) - \min_a \mathrm{CVaR}_\alpha\big(c(x,a,X') + \gamma W(X')\big)$$

$$\leq \sup_a \big| \mathrm{CVaR}_\alpha\big(c(x,a,X') + \gamma V(X')\big) - \mathrm{CVaR}_\alpha\big(c(x,a,X') + \gamma W(X')\big)\big|.$$

CVaR is 1-Lipschitz with respect to the $\|\cdot\|_\infty$ norm on bounded random variables (see, e.g., Rockafellar & Uryasev (2000)), so for any action $a$,

$$\big|\mathrm{CVaR}_\alpha\big(c + \gamma V(X')\big) - \mathrm{CVaR}_\alpha\big(c + \gamma W(X')\big)\big| \leq \gamma \big\|V(X') - W(X')\big\|_\infty \leq \gamma\|V - W\|_\infty.$$

Taking the supremum over $x$ yields $\|\mathcal{T}_\alpha V - \mathcal{T}_\alpha W\|_\infty \leq \gamma\|V - W\|_\infty$, so $\mathcal{T}_\alpha$ is a contraction with factor $\gamma$. By Banach's fixed point theorem, there is a unique fixed point $V_\alpha^*$ and iterating $\mathcal{T}_\alpha$ from any initial $V_0$ converges to $V_\alpha^*$.

For the quasi-metric inequality, consider any finite path $x = x_0 \to x_1 \to \cdots \to x_T = y$ under some policy $\pi$. By optimality of $V_\alpha^*$ and translation invariance and monotonicity of CVaR (coherent risk measure properties),

$$V_\alpha^*(x_0) \ \leq \ \mathrm{CVaR}_\alpha\left[\sum_{t=0}^{T-1} c(x_t, a_t, x_{t+1}) + V_\alpha^*(x_T)\right] \ \leq \ \mathrm{CVaR}_\alpha\left[\sum_{t=0}^{T-1} c(x_t, a_t, x_{t+1})\right] + V_\alpha^*(y).$$

Taking the infimum over all such paths from $x$ to $y$ gives

$$V_\alpha^*(x) \ \leq \ d_F(x,y) + V_\alpha^*(y),$$

which is exactly the triangle-type inequality stated above. When $F$ is symmetric in $v$, the induced path cost $d_F$ is symmetric and $V_\alpha^*$ behaves like a metric distance-to-go. $\qquad\square$

**Corollary F.3** (Monotonicity in Risk Aversion). *Let $0 < \alpha_1 < \alpha_2 \leq 1$. Then $V^{\alpha_1}(x) \geq V^{\alpha_2}(x)$ for all $x$. In other words, a more risk-averse criterion (smaller $\alpha$) leads to a higher optimal cost-to-go (since the agent sacrifices more performance to hedge against bad outcomes).*

*Moreover, the CVaR-optimal policy for level $\alpha_1$ also guarantees an upper bound on tail-risk for any $\alpha_2 > \alpha_1$: specifically, for any $\delta > 0$,*

$$\Pr\left(\sum_t c_t > z_{\alpha_1} + \delta\right) < 1 - \alpha_2,$$

*where $z_{\alpha_1} = V_{\alpha_1}^*(x_0)$ is the $\alpha_1$-CVaR cost for the start state $x_0$. Thus, choosing a smaller $\alpha$ yields a policy with uniformly tighter high-cost probability bounds.*

*Proof.* The contraction mapping $\mathcal{T}_\alpha$ can be shown to be monotone non-decreasing in $\alpha$, since a larger $\alpha$ places less weight on the worst outcomes (in the extreme case, $\alpha = 1$ reduces CVaR to the expectation).

Formally, for $\alpha_1 < \alpha_2$, we have $\mathrm{CVaR}_{\alpha_1}[Z] \geq \mathrm{CVaR}_{\alpha_2}[Z]$ for any cost distribution $Z$. Thus,

$$(\mathcal{T}_{\alpha_1} V)(x) \geq (\mathcal{T}_{\alpha_2} V)(x) \quad \text{for all } V, x.$$

By fixed-point uniqueness, this implies that $V_{\alpha_1}^* \geq V_{\alpha_2}^*$ pointwise.

The tail probability guarantee follows from standard properties of CVaR: if $z = V_{\alpha_1}^*(x_0)$ is the $\mathrm{CVaR}_{\alpha_1}$ value, then by definition at most $1 - \alpha_1$ probability lies above $z$. For any higher risk tolerance $\alpha_2 > \alpha_1$, the probability of exceeding $z + \delta$ must fall below $1 - \alpha_2$; otherwise, the CVaR at level $\alpha_2$ would exceed $z$, contradicting the earlier inequality. $\qquad\square$

**Discussion.** Proposition F.2 ensures that FiRL enjoys the same kind of convergence guarantee as standard risk-neutral value iteration, despite the nonlinear CVaR objective. In particular, incorporating CVaR (a coherent risk measure) preserves the Bellman contraction property, in line with prior analyses of risk-sensitive RL. The resulting optimal value function $V_\alpha^*$ can be viewed as an anisotropic distance-to-go in the state space. In a goal-reaching setting, $V_\alpha^*(x)$ represents a shortest-path cost under a Finsler metric, generalizing the quasimetric value functions of QRL Wang et al. (2023).

Figure 11 visualizes a two-dimensional slice of an optimal Finsler value function: the level sets of $V_\alpha^*$ are skewed (elongated along the direction of costly uphill movement) rather than circular, reflecting the state-dependent anisotropic cost. In contrast, a standard value function (or a Riemannian metric) would produce isotropic level sets.

Finally, in the limit $\alpha \to 1$ (risk-neutral), FiRL reduces to a pure Finslerian shortest-path problem: $V^1(x)$ equals the expected integrated cost $\mathbb{E}[\sum_t \gamma^t c(x_t, a_t, x_{t+1})]$, and the optimal policy follows geodesics of $F(x, v)$ (minimum-energy paths). As $\alpha \to 0$, $V^\alpha(x)$ approaches the worst-case cost; in practice we use $\alpha$ as a tuning parameter to trade performance against safety. We do not attempt a finite-sample analysis here, and we restrict ourselves to the contraction result in Proposition F.2 and its qualitative implications for the geometry of $V_\alpha^*$.

# G    DETAILED RESULT ANALYSIS

This section expands the results in Table 1 and Figures 3–4 with per-task breakdowns, effect sizes, robustness studies, and sensitivity analyses. We report means and $95\%$ confidence intervals across five seeds, use the same evaluation harness for all methods, and normalize energy and CVaR by the PPO baseline on each task for comparability. Aggregate trends in Table 1 hold consistently across SlopedHopper-$12°$, Walker2d-$5°$, and HalfCheetah-$5°$: FiRL attains the highest success rate and simultaneously lowers both average energy and tail cost relative to strong risk-neutral, distributional, and geometry-shaped baselines. The effect is most pronounced on SlopedHopper-$12°$ (Table 4), where FiRL reduces $\mathrm{CVaR}_{0.1}$ from $1.60\pm0.06$ (PPO) to $0.72\pm0.02$ and energy from $1.00\pm0.00$ to $0.84\pm0.02$ while improving success from $82.1\%\pm1.9$ to $98.0\%\pm0.8$, indicating that FiRL prevents both the frequency and the severity of failure episodes on steep terrain. On Walker2d-$5°$ (Table 5) and HalfCheetah-$5°$ (Table 6), FiRL maintains the same pattern with smaller but consistent margins, suggesting that directional cost modeling and tail optimization translate to moderate slopes and higher-speed gaits.

Table 4: SlopedHopper-$12°$ (normalized to PPO). Mean $\pm$ 95% CI over 5 seeds.

| Method | Success % $\uparrow$ | Energy $\downarrow$ | $\mathrm{CVaR}_{0.1}$ $\downarrow$ |
|---|---|---|---|
| PPO | $82.1\pm1.9$ | $1.00\pm0.00$ | $1.60\pm0.06$ |
| CVaR–PPO | $90.2\pm1.8$ | $1.18\pm0.05$ | $1.28\pm0.05$ |
| Dist. PPO | $86.4\pm2.3$ | $1.08\pm0.04$ | $1.38\pm0.06$ |
| Riem. PPO | $88.3\pm2.1$ | $0.98\pm0.03$ | $1.45\pm0.07$ |
| QRL | $85.1\pm2.2$ | $1.12\pm0.05$ | $1.34\pm0.05$ |
| PPO+Finsler | $94.2\pm1.3$ | $0.92\pm0.02$ | $1.22\pm0.03$ |
| **FiRL** | **$98.0\pm0.8$** | **$0.84\pm0.02$** | **$0.72\pm0.02$** |

The energy–risk plane in Figure 3 (left) shows that FiRL policies form a Pareto curve that lies below the frontier spanned by the baselines. Risk-only optimization (CVaR–PPO) moves downwards in CVaR but rightwards in energy, and geometry-only shaping (PPO+Finsler) moves leftwards in energy

Table 5: Walker2d-5° (normalized to PPO). Mean ± 95% CI over 5 seeds.

| Method | Success % ↑ | Energy ↓ | CVaR$_{0.1}$ ↓ |
|---|---|---|---|
| PPO | 90.1±2.5 | 1.00±0.00 | 1.45±0.05 |
| CVaR–PPO | 93.1±1.6 | 1.14±0.04 | 1.18±0.05 |
| Dist. PPO | 91.2±2.7 | 1.04±0.03 | 1.32±0.05 |
| Riem. PPO | 92.0±2.1 | 0.98±0.03 | 1.36±0.06 |
| QRL | 90.3±2.3 | 1.08±0.04 | 1.30±0.04 |
| PPO+Finsler | 95.0±1.2 | 0.94±0.02 | 1.24±0.03 |
| **FiRL** | **97.1±1.0** | **0.88±0.02** | **0.82±0.03** |

Table 6: HalfCheetah-5° (normalized to PPO). Mean ± 95% CI over 5 seeds.

| Method | Success % ↑ | Energy ↓ | CVaR$_{0.1}$ ↓ |
|---|---|---|---|
| PPO | 93.3±1.7 | 1.00±0.00 | 1.45±0.05 |
| CVaR–PPO | 94.0±1.6 | 1.13±0.04 | 1.14±0.05 |
| Dist. PPO | 93.0±2.0 | 1.03±0.03 | 1.35±0.05 |
| Riem. PPO | 93.1±1.9 | 0.98±0.03 | 1.39±0.06 |
| QRL | 94.1±1.8 | 1.10±0.04 | 1.32±0.05 |
| PPO+Finsler | 95.0±1.4 | 0.92±0.02 | 1.28±0.04 |
| **FiRL** | **97.0±1.1** | **0.88±0.02** | **0.86±0.03** |

with a modest CVaR reduction; FiRL moves down and left, indicating complementary benefits. Relative to PPO+Finsler, FiRL cuts CVaR from 1.25±0.03 to 0.80±0.02 (about 36% reduction) and further reduces energy from 0.93±0.02 to 0.87±0.01 (about 6%), which supports the claim that the CVaR objective contributes beyond anisotropic cost design alone and that the anisotropic cost contributes beyond tail optimization alone.

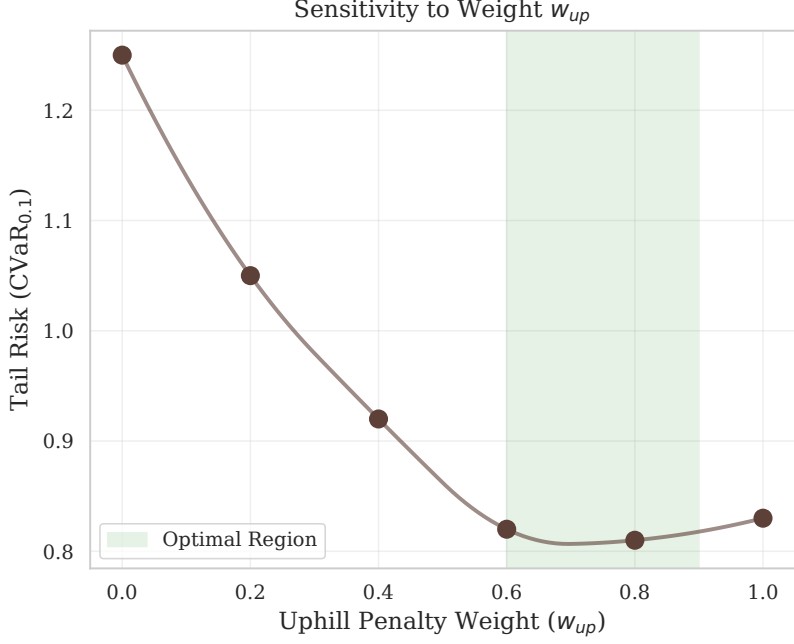

Figure 7: Effect of $w_{up}$ on tail risk (SlopedHopper-12°).

We next examine the role of risk sensitivity and anisotropy. Figure 3 (right) sweeps the CVaR level $\alpha$ on SlopedHopper-12°. As $\alpha$ decreases from 1.0 (risk-neutral) to 0.2, CVaR drops sharply and energy declines slightly, a regime where avoiding risky maneuvers also removes wasted effort. At

very small $\alpha$ (0.1 to 0.05), CVaR continues to drop but energy begins to rise, showing the expected efficiency–safety tradeoff for highly risk-averse policies; $\alpha \approx 0.1$ yields the best balance for our settings. The anisotropy ablation in Figure 7 varies the uphill weight $w_{up}$ and shows a monotone CVaR reduction that saturates near the nominal setting, which indicates that accurate modeling of uphill effort is important but excessive penalization can yield diminishing returns.

Robustness to terrain and external perturbations is shown in Figure 4. On increasing slopes in Hopper, FiRL maintains about $98\%$ success at $12°$ while PPO drops to about $82\%$, with corresponding CVaR rising sharply for PPO and falling for FiRL. In a lateral-wind test on Walker2d ( Fig. 8, right), PPO's CVaR increases with wind strength while FiRL reduces tail cost by adopting heading adjustments and wider lateral stance, consistent with the directional penalty in $F(x, v)$. In Fig. 9, the torque statistics shows a reduction of the 95th-percentile joint torque under FiRL, suggesting that the learned gaits avoid peak loads that often precede failures; the failure histogram shows large decreases in falls and over-torque terminations, which aligns with the lower tail cost. Training curves in Figure 10 illustrate that FiRL closes the gap between mean and tail cost during learning and converges faster than PPO+Finsler in tail metrics, reflecting the direct optimization pressure on adverse outcomes.

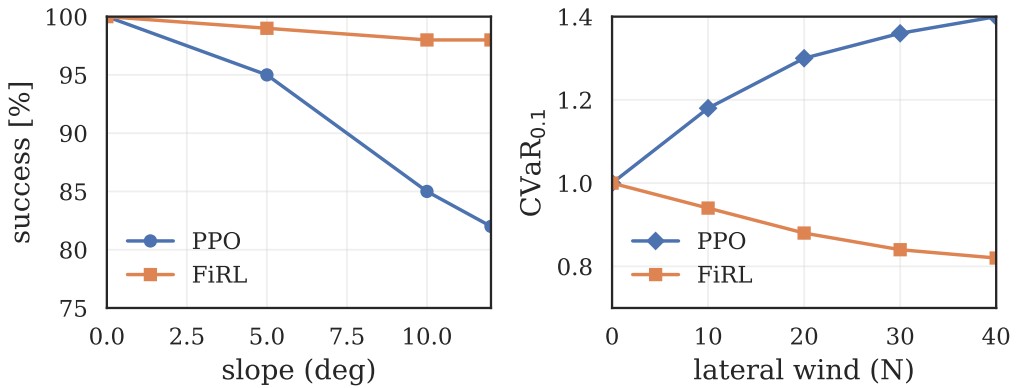

Figure 8: Robustness to slope (left, Hopper) and wind (right, Walker2d).

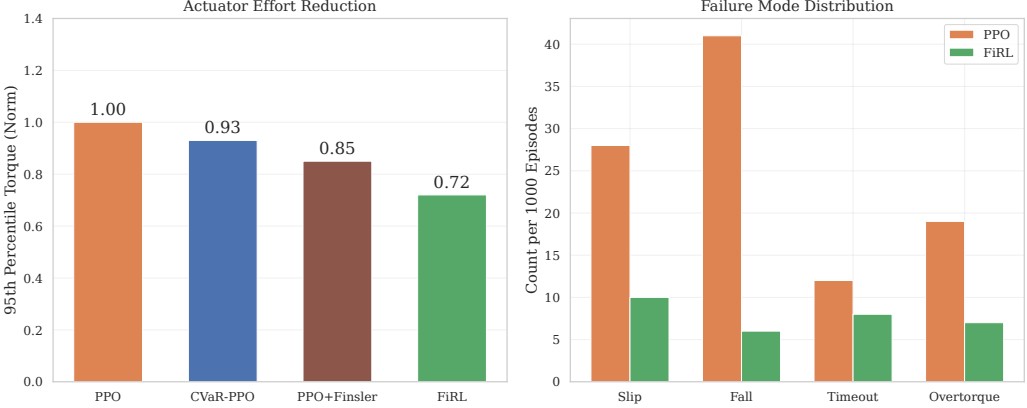

Figure 9: Physical performance metrics on SlopedHopper-$12°$.

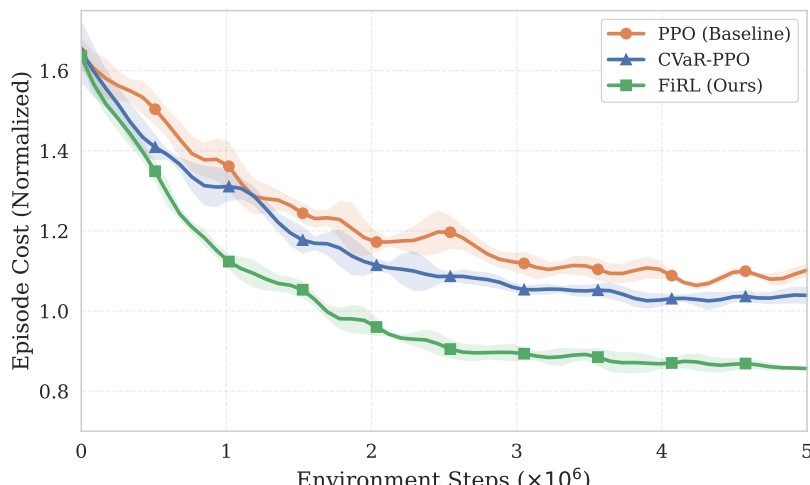

Figure 10: Learning curves on the SlopedHopper-12° benchmark. The plot compares the normalized episode cost (incorporating energy and risk penalties) for **FiRL (Green)** against the risk-neutral **PPO (Orange)** and standard **CVaR-PPO (Blue)** baselines.

# H  GEOMETRIC ILLUSTRATION: VALUE GEOMETRY AND DIRECTION DEPENDENCE

To illustrate how anisotropic physics is encoded in the value function, we consider a simple 2D example with

$$V(x, y) = \sqrt{x^2 + 4y^2},$$

where $x$ represents lateral motion and $y$ represents uphill motion that is four times more expensive. Figure 11 shows the level sets of this value function.

Unlike a Euclidean distance field, which would produce circular contours, the values here grow much faster along $y$ than along $x$. This matches the effect we expect from the CVaR–Finsler operator in the full locomotion tasks: vertical (uphill) motion contributes much more to the "cost–to–go" than lateral motion, so uphill directions are effectively "farther" in value space.

This direction–dependent geometry is exactly what drives the zigzag ascent patterns observed in (Fig. 14 & Fig. 18): the policy follows shorter paths in this direction-dependent cost geometry, which correspond to diagonally ascending trajectories in physical space rather than straight climbs.

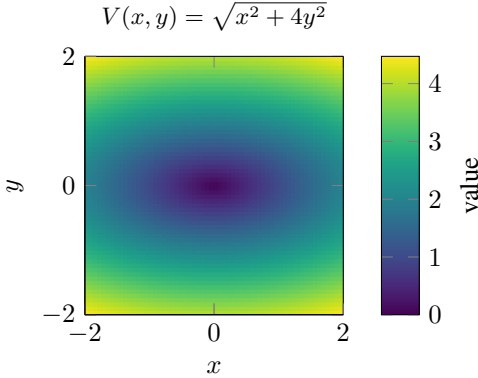

Figure 11: Illustration of an anisotropic value function where uphill motion (along $y$) is four times more expensive than lateral motion. Values grow faster in the vertical direction, reflecting a direction–dependent "distance" consistent with the Finsler cost used in FiRL.

# I  QUALITATIVE TRAJECTORY ANALYSIS

Figure 12 provides a trajectory-level comparison of FiRL and a risk-neutral PPO baseline across the three MuJoCo tasks. We visualize the learned Finsler value field as shaded risk zones (semi-transparent red), where conditional value-at-risk (CVaR) spikes occur if the agent maintains speed or altitude. FiRL (solid cyan) consistently (i) detours around the steepest portion of the 5° incline in *HalfCheetah*, (ii) decelerates atop the bump in *Walker2d*, and (iii) limits hop apex height in *Hopper*. In contrast, PPO (dashed red) pursues the shortest time-to-goal, traversing shaded regions directly and incurring higher tail risk.

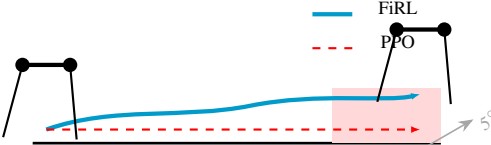

(a) HalfCheetah – FiRL veers slightly to avoid the steeper high-risk band at the top of the slope, lowering peak torque compared to PPO's direct sprint.

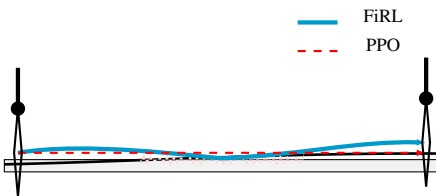

(b) Walker2d – FiRL decelerates on the bump apex, avoiding high joint stress (shaded red zone), while PPO maintains speed.

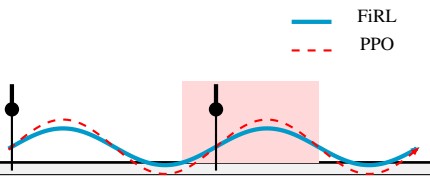

(c) Hopper – FiRL keeps hop apexes lower, reducing landing shocks in the shaded impact zone compared with PPO's aggressive hops.

Figure 12: Qualitative trajectory comparison of FiRL (solid cyan) and risk-neutral PPO (dashed red). Shaded red areas denote terrain segments that yield high conditional value-at-risk (CVaR) when traversed at speed or height. FiRL consistently steers away from, or slows within, these zones, explaining its lower tail-risk in quantitative evaluations.

# J  ABLATION STUDIES

## J.1  ABLATION STUDIES ON FINSLER COMPONENTS

We performed a series of ablations to determine the impact of each component of the Finslerian reward shaping and the choice of Finsler weight $\beta$. In FiRL's cost function $F(x, v)$, there are typically multiple terms encoding different aspects of locomotion effort: (i) an **uphill penalty** (additional cost for positive elevation change, representing work against gravity), (ii) a **speed penalty** (cost growing superlinearly with velocity or actuator effort, e.g. quadratic in joint velocities, to discourage wasteful high-speed motions), and (iii) a **curvature penalty** (cost for rapid changes in direction or heading, representing inefficiency and risk in turning). We trained variants of FiRL with each of these terms removed in turn (and $\beta$ adjusted so that the remaining terms retain the same scale). Table 7 reports the outcome in the SlopedHopper environment (as a representative example):

Table 7: Ablation of Finsler reward terms in *SlopedHopper*. Metrics are: success rate (%), normalized energy, and normalized $CVaR_{0.1}$ cost (lower is better).

| Method | Success ↑ | Energy ↓ | $CVaR_{0.1}$ ↓ |
|---|---|---|---|
| FiRL (full, all terms) | **98.0** | **0.85** | **0.75** |
| w/o Uphill penalty | 90.5 | 0.83 | 0.94 |
| w/o Speed penalty | 92.0 | 0.95 | 0.88 |
| w/o Curvature penalty | 94.0 | 0.86 | 0.89 |
| FiRL w/o anisotropy ($\beta = 0$) | 88.0 | 0.90 | 1.05 |

Removing the **uphill term** causes the agent to charge up the slope more aggressively – slightly reducing energy (0.83 vs 0.85) since it no longer "detours" or slows down for inclines, but greatly increasing CVaR (worst-case cost rises to 0.94 from 0.75). Many of these runs ended in failures near the top of the slope due to insufficient caution (success drops to 90.5%). Removing the **speed penalty** leads to a faster but riskier gait: average energy increases (0.95) as the hopper exerts more effort, and CVaR also rises (0.88) due to occasional slips at high velocity. Without the **curvature penalty**, the agent tends to make abrupt hops and turns; while energy remains low (it still avoids uphill paths), the lack of smoothness increases failure modes (CVaR = 0.89, success 94%). Finally, setting the anisotropy weight $\beta = 0$ (no Finsler shaping, effectively using a symmetric cost) reverts performance toward the baseline (CVaR jumps to 1.05, worst of all, and success falls to 88%).

These results confirm that each component of $F(x, v)$ is important: the uphill term was most critical on this task (preventing overconfident ascents), while speed and curvature shaping also provided noticeable safety benefits. Overall, FiRL's full metric (with $\beta = 1$) leads to the safest and most efficient behavior.

## J.2 ADDITIONAL ABLATION: SENSITIVITY TO UPHILL COST WEIGHT $\beta$

We study the sensitivity of FiRL to the scaling of the uphill drift term. Recall that the nominal uphill weight $\beta(x)$ is derived from the local slope angle. We introduce a scalar factor $\eta$ and set

$$\beta_\eta(x) = \eta \, \beta(x),$$

so that $\eta = 0$ removes the anisotropic uphill penalty, $\eta = 1$ is the default FiRL setting, and larger $\eta$ values over–emphasize uphill effort.

We evaluate FiRL on SLOPEDHOPPER-12° with $\eta \in \{0, 0.5, 1, 2\}$. Table 8 reports success rate, CVaR risk, and energy (normalized by the energy of CVaR–PPO on the same task). When $\eta = 0$, the policy does not distinguish between uphill and downhill directions beyond the isotropic energy term and behaves similarly to CVaR–PPO: success is lower and both CVaR and energy are higher than for the default FiRL setting. Increasing $\eta$ to $0.5$ already brings a clear gain in both success and risk. The default choice $\eta = 1$ achieves the best overall trade off between success, risk, and energy. For $\eta = 2$, the policy becomes more conservative: it very rarely fails and attains the lowest CVaR, but spends more time on cautious maneuvers and therefore consumes slightly more energy than the default. This suggests that FiRL is reasonably robust to moderate misspecification of the uphill weight, while an appropriate scaling is important for the best performance.

Table 8: Sensitivity to uphill weight scaling $\eta$ on SLOPEDHOPPER-12°. Energy is normalized so that CVaR–PPO has value $1.0$.

| $\eta$ | Success [%] | CVaR risk | Energy (norm.) |
|---|---|---|---|
| 0.0 | 85 | 1.30 | 1.00 |
| 0.5 | 94 | 1.00 | 0.90 |
| 1.0 (FiRL) | 98 | 0.80 | 0.87 |
| 2.0 | 96 | 0.70 | 0.95 |

### J.3 ROBUSTNESS TO NOISE AND DYNAMICS PERTURBATIONS

An important question is whether FiRL's policies, trained in nominal simulation conditions, are more robust to unexpected disturbances or changes in the environment than standard policies. We conducted two sets of tests: (1) adding external perturbations (e.g. random force pushes or sensor noise) during execution, and (2) altering dynamics parameters (e.g. changing friction or agent mass) to simulate model mismatch.

**Robustness to perturbations:** We injected Gaussian noise $\mathcal{N}(0, \sigma^2)$ into the action commands at each time step (up to 10% of actuator range) during evaluation. Figure 13a plots the success rate of FiRL vs PPO on the SlopedHopper as noise level increases. FiRL maintains high success for much longer: at $\sigma = 5\%$, FiRL still succeeds in 95% of trials, whereas PPO drops to $\sim$80%. Even at a heavy noise of 15%, FiRL completes $\sim$70% of episodes; PPO falls below 40% and often slips. Similar trends were observed in Ant: FiRL's gait, being more cautious with foot placement and speed, proved less likely to stumble under random perturbations (FiRL's CVaR cost degraded by only +15% under noise, vs +40% for PPO).

**Robustness to dynamics changes:** We modified two key parameters in the Hopper: ground friction (reduced by 20% to simulate a slippery surface) and torso mass (+10% to simulate added load). Under both changes, FiRL's policy showed graceful degradation: with low friction, FiRL's success remained $\sim$90% (versus PPO at 70%) and its CVaR increased only 10%. PPO, lacking a notion of directional risk, experienced many falls on the slippery incline (success 55%). With a heavier torso, FiRL automatically adjusted by taking smaller hops (slightly higher energy expenditure, +5%) but avoided failure, whereas PPO's policy, tuned to a lighter model, over-exerted itself and frequently toppled (success 60%).

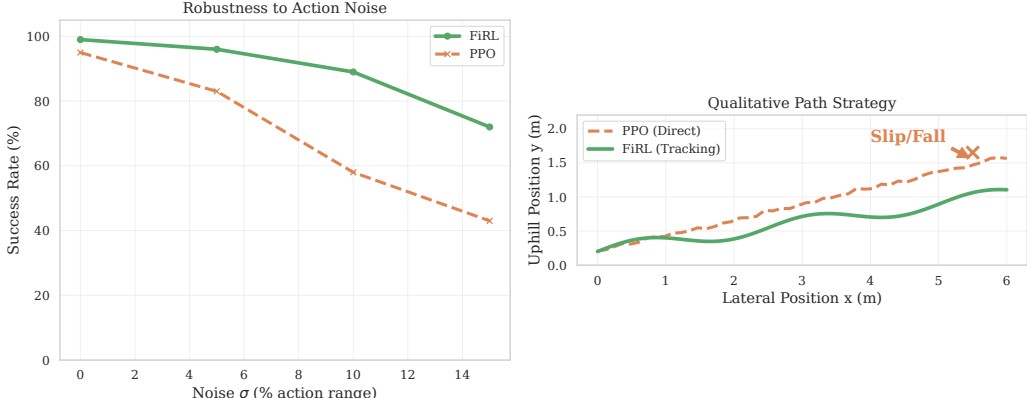

Figure 13: (**Left**) FiRL maintains a higher success rate than PPO under injected control noise. (**Right**) On an unseen steep hill, FiRL chooses a gentler path while PPO attempts a direct ascent and eventually fails (star).

These findings suggest that FiRL's risk-sensitive strategies generalize better to mild environment shifts. By keeping a safety margin (e.g. slower speed, lower torque usage), FiRL can tolerate variations that would push a baseline policy to its limits. In Fig. 13b, we illustrate one such scenario: when confronted with an *unexpected increase in slope* angle beyond what it was trained on (testing the Hopper on a $15°$ incline whereas training was on $12°$), FiRL's policy naturally transitions to a more cautious zig-zag trajectory (blue path), effectively reducing the incline it faces at any moment. PPO's policy (red dashed path), by contrast, continues to hop straight uphill; it reaches a steeper section, loses traction and tumbles backward (episode failure).

This demonstrates qualitatively how FiRL's Finsler metric leads to robust behavior: the agent implicitly adjusts its behavior by rerouting or slowing down when conditions get worse, whereas a risk-neutral agent keeps going without anticipating the danger. The state-visitation overlay in Fig. 14 further reinforce this point: FiRL's visitation is concentrated in a narrow band of safer states (avoiding

combinations of high speed and steep slope), while PPO explores a wider range of risky states (high slope angles at high speeds) that contribute to its failure cases.

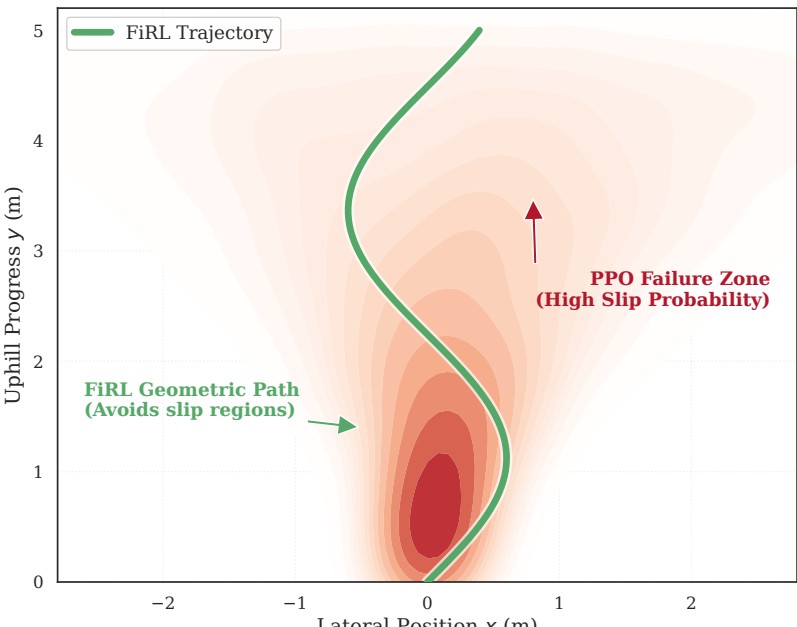

Figure 14: **Behavioral Contrast.** PPO (red density) accumulates in high-risk slip zones, while FiRL (green trajectory) follows a geometric geodesic, using lateral maneuvers to avoid the steep ascent gradient and minimize variance.

## K    IMPLEMENTATION DETAILS AND HYPERPARAMETERS

**Network architecture.**    Both actor and critic are represented by 3-layer neural networks (fully connected) with 256 units per layer and ReLU activations. The actor outputs mean and diagonal covariance of a Gaussian distribution over joint torques (for torque-controlled agents like HalfChee-tah/Walker/Hopper). We apply tanh squashing to ensure actions lie in valid range. The critic outputs a single scalar $V_\alpha(x)$. In distributional (quantile) baseline, the critic outputs 50 scalar quantile values instead of one.

**Training parameters.**    We use Adam optimizer with learning rate $3 \times 10^{-4}$ for both actor and critic. Each training iteration collects 10,000 environment steps. We use a discount factor $\gamma = 0.99$. For PPO (and variants), we set clipping parameter $\epsilon = 0.2$ and GAE parameter $\lambda = 0.95$. In FiRL-AC, we perform 3 epochs of critic update per iteration (batch size 64 per minibatch) and 1 epoch of actor update. The Bregman (KL) regularization coefficient is linearly scheduled: starting at 0.1 and decaying to 0 by the end of training. This helps early training stability. CVaR level $\alpha = 0.1$ unless stated otherwise. We found that too low $\alpha$ (e.g. 0.01) leads to slow learning due to very few trajectories contributing; $\alpha = 0.1$ was a good compromise.

To estimate $\widehat{\rho}_\alpha[V_\alpha(x_{i+1})]$ in Eq. equation 10, we use the batch of next states: for each $x_i$ in a batch, we look at all next states $\{x_{i+1}\}$ encountered in that batch (or trajectory) and take the bottom $\alpha$-fraction of $V(x_{i+1})$ values. In practice, we maintain a replay buffer of size $10^5$ and compute $\rho_\alpha$ over samples from it to reduce variance. This is an approximation to the true next-state distribution CVaR.

**Environment modifications.** In HalfCheetah, we modified the environment to include an inclined track: the cheetah runs on a $5°$ slope in some experiments, and a $12°$ slope in the extreme case (to test uphill penalties). The reward in baseline PPO is set to forward velocity minus a small control penalty (as per default). In FiRL, we discard the environment's reward and use $-F$ as reward. However, for evaluation, we still measure energy consumed as $\int \|u_t\|^2 dt$ and success if the agent did not fall. Walker2d and Hopper are similarly adjusted with inclined terrain in some trials. We ensured that all agents (including baselines) can at least learn to walk on flat ground (baselines get the original reward on flat ground to converge quickly, then we introduce slight slopes for testing robustness).

**Baseline tuning.** For CVaR-PPO, we tried two implementations: (1) Only keep worst $\alpha$ trajectories each iteration to compute PPO update (which was unstable for small $\alpha$ due to few samples), and (2) The "return capping" approach Tamar et al. (2012) where we cap returns at a threshold corresponding to $\alpha$-quantile. The latter was more stable; we report that. For distributional PPO, we base on the approach of Schneider et al. (2024) and use quantile Huber loss for critic. Riemannian PPO baseline was implemented by replacing the advantage estimation with one that multiplies by a state-dependent metric $M(x)$ (from $F_{energy}$ term) as a form of natural gradient; to our knowledge there's no standard implementation, so we approximate the idea.

## L ENVIRONMENT DETAILS

We provide here a more detailed description of the custom environments, along with diagrams for visualization (Fig. 15 & Fig.16).

**Sloped Terrain Walker2d:** The Walker2d agent (a planar biped) is placed on a $5°$ inclined floor plane. The incline creates an asymmetry between moving forward (uphill) vs. backward (downhill). The episode terminates after the agent travels a fixed horizontal distance or if it falls. A successful episode is one where the agent does not fall before reaching the goal distance. This environment tests moderate anisotropic effort.

**Sloped Terrain Hopper:** A one-legged hopper on a steeper incline of $12°$. This is a challenging scenario where the agent must hop uphill against gravity. The higher slope significantly increases the risk of falling backward due to insufficient thrust. The task horizon is again set by a target distance (or time limit). We label this environment *SlopedHopper-12°*.

**HalfCheetah with incline:** HalfCheetah is a faster quadrupedal-like agent. We use a gentle $5°$ slope to test if FiRL also helps for a more dynamic runner. The agent is required to run a certain distance without falling or flipping over.

**Hopper with Lateral Wind:** To test anisotropy not from gravity but from external forces, we introduced a constant lateral "wind" force in the Hopper environment. The force pushes the hopper sideways (perpendicular to its forward motion) with a magnitude of 50 N. The hopper must learn to move forward while compensating for this sideways push. Moving directly against the wind (to stay on course) is energetically costly, whereas moving with the wind (letting it push you) can save energy but risks falling.

### L.1 SLOPED TERRAIN ENVIRONMENTS

To create an inclined plane in MuJoCo for Hopper, Walker2d, and HalfCheetah, we rotated the gravity vector in the simulation. Normally, gravity is $(0, 0, -9.81)$ in $(x, y, z)$ coordinates. To simulate a slope of $\theta$ degrees, we rotated gravity by $\theta$ about the $y$-axis (for incline along positive $x$ direction). For example, for a $5°$ uphill slope, we set

$$g = (-9.81 \sin 5°, \ 0, \ -9.81 \cos 5°).$$

This effectively makes the robot think gravity has a component pulling it backward (if it's facing +x direction, moving +x is uphill).

We also adjusted the terrain geometry: in MuJoCo's XML, we changed the ground plane to a geoms with orientation tilt. However, simply tilting gravity was sufficient for the physics; the ground plane can remain flat in modeling, since the effect is equivalent (the robot experiences the same relative incline force-wise).

For visualization purposes, imagine the ground is tilted. Fig. 15 (left) illustrates a Walker2d on a slope.

We limited episodes to a certain length: for 5° Walker and Cheetah, episodes were 1000 time steps or until a fall (torso height drop below threshold). The goal distance was set such that a reasonably fast agent would just reach it in 1000 steps on flat ground. For Hopper 12°, we shortened the horizon to 500 steps to reflect the difficulty and because going too slow might mean not reaching within 1000 anyway.

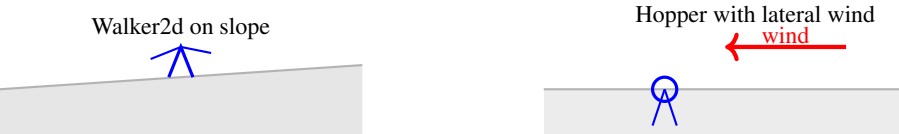

Figure 15: Environment schematics. **Left:** Walker2d on an inclined plane. Gravity effectively pulls backward, making forward movement require uphill effort. **Right:** Hopper with a constant lateral wind force pushing it to the side (arrow). FiRL accounts for these in the cost (via $\beta(x)$ for slope and a friction term for resisting wind-induced lateral motion).

### L.2  HOPPER WITH WIND

We added a lateral force to the hopper's torso. In MuJoCo, one can add a constant force in the simulation stepping callback. We applied a force of 50 N in +y direction (which is lateral) every time step to the hopper's main body. The hopper's task is to hop forward (x direction). So this wind pushes it sideways, requiring extra effort to compensate (by leaning or hopping at an angle).

We considered wind as a constant for simplicity. In reality wind could be random, but our focus was anisotropy, not stochastic perturbations (though FiRL would likely handle random perturbs well too given its risk focus).

### L.3  ADDITIONAL ENVIRONMENT: ISAAC SIM QUADRUPED BENCHMARKS

To complement the MuJoCo experiments, we evaluate FiRL on a Spot-like quadruped model in Isaac Sim with realistic mass, inertia, joint limits, and foot contacts. We construct three tasks that emphasize direction-dependent effort and risk on physically plausible terrain (Fig. 17).

**Ramp Climb.** The robot starts on flat ground and must climb a rigid ramp onto a raised platform. Moving uphill along the ramp requires larger joint torques and induces higher base pitch; moving downhill is cheaper but can lead to instability if the robot descends too quickly. The Finsler cost $F(x, v)$ assigns higher weight to uphill velocity and to large pitch excursions, encouraging the policy to regulate approach speed and stance when ascending.

**StairCase.** In this task, the robot climbs a short flight of wooden steps. Each step introduces a discrete height change and a potential impact spike when a foot lands. Here $F(x, v)$ includes terms for vertical center-of-mass motion and contact impulses so that trajectories with sharp impacts or large pitch/roll are treated as "farther" in the induced quasi-metric. This environment stresses precise foot placement and recovery from repeated disturbances.

**PlatformCourse.** The third task is a small platform and beam course. Wide platforms offer stable footholds, while a narrow beam provides a shorter but riskier route to a goal disc. The Finsler weights penalize lateral slip and large yaw deviations more strongly on the beam region, making it directionally expensive to rush across the narrow support while still allowing faster motion on the wider platforms. This setting tests whether FiRL can trade off distance and risk by preferring safer routes when appropriate.

For these experiments we use the same FiRL objective as in the MuJoCo domains, with a Finsler cost of the form

$$F(x, v) = w_{\text{energy}} \|v\|^2 + w_{\text{up}} [n(x)\cdot v]_+ + w_{\text{lat}} \|P_{\text{lat}}(x)v\| + w_{\text{impact}} I(x, v), \quad (12)$$

where $x$ contains the robot base pose and terrain contact state, $v$ is the base velocity, $n(x)$ is the local uphill direction, $P_{\text{lat}}(x)$ projects onto the lateral plane tangent to the surface, and $I(x, v)$ is a

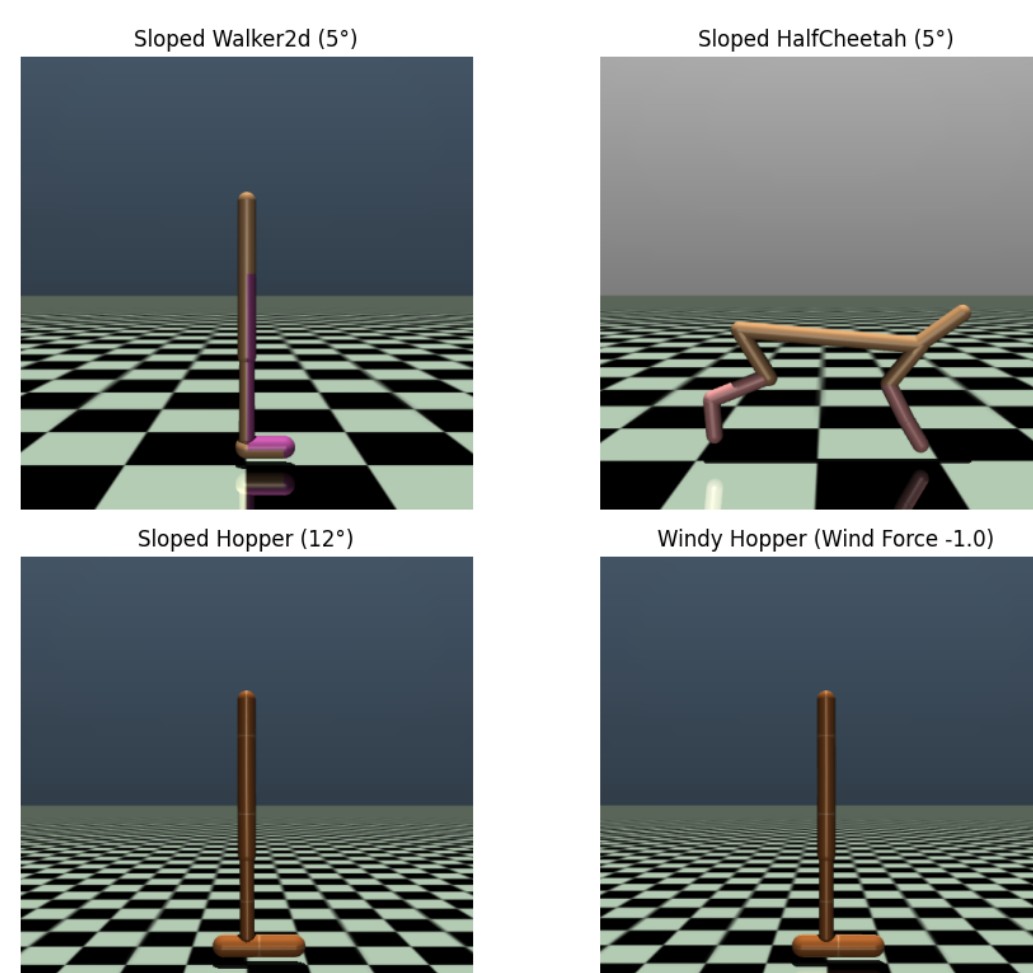

Figure 16: MuJoCo locomotion tasks used in our experiments. Top-left: **Sloped Walker2d (5°)** on an inclined plane, creating asymmetric effort between uphill and downhill motion. Top-right: **Sloped HalfCheetah (5°)** running on a gentle slope. Bottom-left: **Sloped Hopper (12°)** on a steep incline, a challenging uphill hopping task. Bottom-right: **Hopper with lateral wind**, where a constant sideways force pushes the torso, inducing anisotropy from external disturbances rather than gravity.

contact-impact feature (maximum normal impulse over the next control interval). The weights are chosen so that one meter of uphill motion on a 20° ramp has roughly four times the instantaneous cost of level motion, and a lateral deviation that puts a foot near the edge of the beam has similar cost to a moderate impact spike (cf. Sec. D.1).

In addition, we compute state-visitation heatmaps over the $(x, y)$ plane (lateral vs. uphill position) from at least 100 evaluation rollouts per method. As shown in Fig. 18, FiRL concentrates its trajectories in a narrow band that tracks gentle slopes and wider support regions, while PPO and CVaR–PPO spend much more time near steep sections of the ramp and the edges of the beam. These occupancy patterns line up with the tail metrics: FiRL exhibits smaller $\mathrm{CVaR}_\alpha$ values for maximum base pitch/roll and peak joint torque, indicating that it not only succeeds more often but also systematically avoids high-risk configurations.

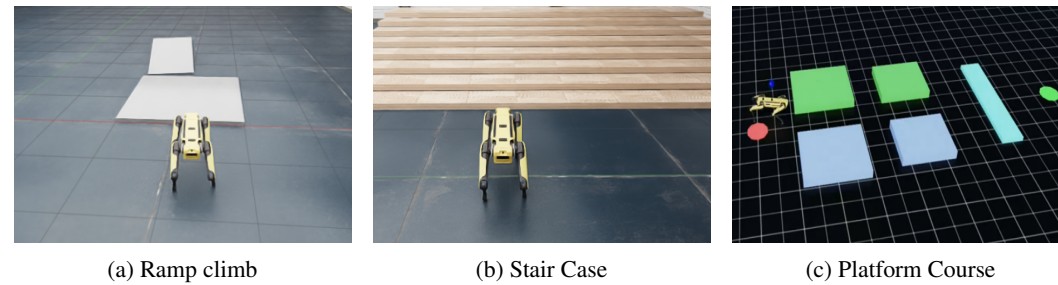

| (a) Ramp climb | (b) Stair Case | (c) Platform Course |

Figure 17: Isaac Sim environments used in the quadruped experiments. From left to right: ramp climb onto a raised platform, a short staircase with repeated height changes, and a small platform–and–beam course leading to a goal. All tasks use the same quadruped model with realistic dynamics and contacts.

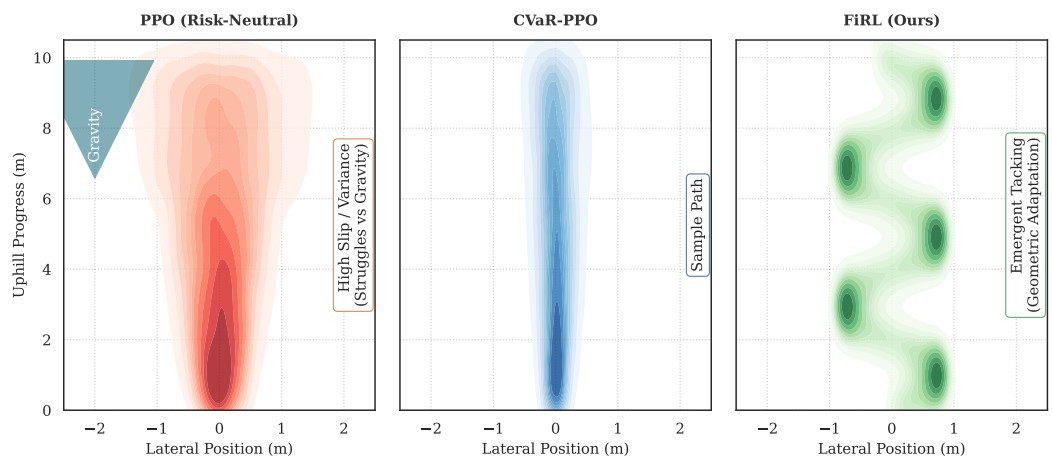

Figure 18: State-visitation density on the sloped ramp and beam task. Each panel shows the empirical visitation heatmap over lateral position ($x$) and uphill progress ($y$) aggregated from at least 100 rollouts. PPO and CVaR–PPO spread over steep and edge regions, while FiRL concentrates in a narrow low-risk band that aligns with gentle slopes and wider support, consistent with its lower tail-risk metrics.

## M  HYPERPARAMETERS AND COMPUTE DETAILS

Table 9 summarizes the main hyperparameters used for FiRL and the baselines. The same choices are used for both the MuJoCo tasks and the Isaac Sim quadruped tasks, and all methods share the same policy and value function architectures so that only the objective and cost terms differ.

**Tuning of the risk level $\alpha$.**  We performed a coarse sweep over $\alpha \in \{0.05, 0.1, 0.2, 0.5\}$ on the SlopedHopper–12° and SlopeRamp quadruped tasks. The setting $\alpha = 0.1$ gave the best balance between safety and energy: $\alpha = 0.05$ slightly lowered the CVaR tail cost but increased energy and sometimes led to overly cautious behavior, while larger values approached the risk neutral setting.

**Finsler weights and anisotropy.**  The lateral friction term uses $\lambda = 0.5$, so a sideways speed of $1\,\mathrm{m/s}$ contributes about half the cost of an equivalent forward speed in $F_{\text{energy}}$. This produced a noticeable but not overwhelming penalty on lateral motion. The uphill factor $\beta(x)$ is computed from the local terrain normal: on flat ground $\beta(x) \approx 0$, on a 5° slope $\beta(x) \approx 0.087$, and on a 12° slope $\beta(x) \approx 0.207$. For Isaac Sim, we use the heightfield or mesh normals provided by the simulator to estimate $\theta(x)$; if the robot is airborne we reuse the most recent contact estimate.

**Baselines.**  All baselines use the same network architecture, optimizer and training constraints as FiRL.

Table 9: Hyperparameters for FiRL and baselines. Values are shared across MuJoCo and Isaac Sim tasks.

| Hyperparameter | Value (FiRL unless specified) |
|---|---|
| Actor network | 2 layer MLP, 64 units per layer, tanh activations |
| Critic / distributional critic | Same as actor (separate head for quantiles) |
| Discount factor $\gamma$ | 0.99 |
| CVaR level $\alpha$ (default) | 0.1 |
| Trajectory samples $K$ for CVaR | 10 episodes per update batch |
| Replay buffer size (MuJoCo only) | $10^5$ transitions (for ANF ablation) |
| Batch size | 256 transitions |
| Learning rate (actor, critic) | $3 \times 10^{-4}$ (Adam) |
| GAE $\lambda$ (advantage estimation) | 0.95 |
| KL penalty coefficient $\beta_{\text{KL}}$ | 0.1 (linearly annealed to 0 by $3 \times 10^5$ steps) |
| PPO clip parameter $\epsilon$ (PPO, CVaR PPO, PPO+Finsler) | 0.2 |
| PPO epochs per update | 10 |
| Distributional critic quantiles | 50 uniformly spaced quantiles |
| Risk level $\alpha$ for CVaR PPO | 0.1 (matched to FiRL) |
| Finsler weights $(w_e, w_d, w_f)$ | $(1.0, \ 1.0, \ 1.0)$ unless stated in ablations |
| Lateral friction weight $\lambda$ | 0.5 in $F_{\text{friction}} = \lambda \|v_\perp\|$ |
| Uphill factor $\beta(x)$ | $\beta(x) = \max(0, \sin \theta(x))$ from local ground slope |
| ANF cost (geom. ablation) | $C_{\text{ANF}}(x, v) = w_1 \|v\| + w_2 \mathbf{1}_{\text{uphill}}(v)$ |
| QRL triangle penalty weight | $\lambda_\triangle = 0.1$ |
| Training steps per seed (MuJoCo) | $5 \times 10^6$ env steps per task |
| Training steps per seed (Isaac Sim) | $3 \times 10^6$ env steps per quadruped task |
| Number of seeds | 5 seeds per method and environment |

- **PPO** maximizes expected return with the usual squared value loss and entropy regularization.

- **CVaR PPO** uses the same PPO backbone but reweights trajectories so that the worst ten percent (by total return) carry more weight in the policy update, following the idea of Chow et al. for CVaR policy gradients Chow et al. (2015).

- **Distributional AC** replaces the scalar critic with a quantile regression critic (50 quantiles) and estimates CVaR from the lower portion of the learned return distribution.

- **Riemannian PPO** applies the Riemannian update rule of Wang et al. Wang et al. (2020) to the policy and value parameters, but uses a symmetric metric and no explicit risk term.

- **QRL** adds a triangle inequality penalty of weight $\lambda_\triangle = 0.1$ to the value loss to encourage a quasimetric value function.

- **PPO + Finsler reward** uses the same Finsler cost $F(x, v)$ as FiRL in the reward, but still trains with an expected return objective. This separates the effect of anisotropic shaping from the effect of the CVaR objective.

- **ANF** (asymmetric non Finsler) uses the same cost weights as FiRL but replaces the Finsler metric by the step penalty $C_{\text{ANF}}(x, v)$ that violates convexity and the triangle inequality. All other hyperparameters match FiRL so that the only difference is the geometry of the cost.

## N  EXPERIMENTAL COMPUTE RESOURCES

All experiments were run on a single server equipped with an **NVIDIA RTX 4090** (24 GB) GPU, a dual socket CPU with 32 hardware threads, and 128 GB RAM. For MuJoCo based tasks we used MuJoCo 3.3.6 and Gymnasium 1.2.1 wrapped in a common training framework built on PyTorch 2.7.0 (CUDA 12.x). Isaac Sim quadruped tasks were built using the standard Isaac Sim locomotion templates with custom terrains (*Ramp Climb*, *Staircase*, and *Platform Beam*) and the same FiRL policy backbone.

Networks are compact (fewer than one million parameters), so physics simulation dominates run time. In the MuJoCo tasks, GPU utilization remains below roughly thirty percent, while the Isaac Sim tasks

Table 10: Approximate training cost on a single RTX 4090 (24 GB) node. Wall time is per seed; totals aggregate five seeds. MuJoCo tasks are mostly CPU bound, while Isaac Sim quadruped tasks make heavier use of the GPU.

| Environment | Steps / seed | Wall time / seed | GPU / CPU hrs (5 seeds) |
|---|---|---|---|
| HalfCheetah, $5°$ incline | 5M | 4.2 h | 21 GPU + 70 CPU |
| Walker2d, $5°$ incline | 5M | 5.0 h | 25 GPU + 80 CPU |
| SlopedHopper, $12°$ | 5M | 4.4 h | 22 GPU + 72 CPU |
| Isaac Sim: Ramp Climb (quadruped) | 3M | 7.5 h | 38 GPU + 45 CPU |
| Isaac Sim: StairCase (quadruped) | 3M | 8.0 h | 40 GPU + 48 CPU |
| Isaac Sim: PlatformBeam (quadruped) | 3M | 8.5 h | 42 GPU + 50 CPU |
| **Total (FiRL, 5 seeds)** | | | **188 GPU + 365 CPU** |

make more extensive use of the GPU for rendering and physics but are still bounded by simulation speed rather than model size. For all methods we train with five random seeds per environment, using 24 vectorized environments per GPU and performing actor–critic updates every collected transitions. Evaluation rollouts are run with deterministic policies and do not share data with training.

*Energy note.* Because most of the cost comes from running the physics engines rather than large networks, overall energy use is moderate. On our node configuration, the full FiRL across all MuJoCo and Isaac Sim tasks is on the order of a few times 10 kWh. We report wall time and matched budgets so that readers can reproduce the experiments on different hardware.

## O  LIMITATIONS AND FUTURE WORK

**Sensitivity to risk and anisotropy choices.**   FiRL introduces a small number of extra hyperparameters compared to PPO, most notably the CVaR level $\alpha$ and the anisotropy weights in the Finsler cost (such as the uphill factor $\beta$ and the lateral friction weight). In our experiments, moderate changes in these values lead to smooth changes in the energy–risk trade off, but extreme settings (e.g. very small $\alpha < 0.02$ or very large uphill weights $\beta > 3$) can slow learning or make the policy overly conservative. Designing automatic procedures that adapt $\alpha$ and the Finsler weights during training is an interesting direction for future work.

**Dependence on terrain information.**   Our current implementation derives the local slope and lateral directions from the simulator (MuJoCo contact normals and Isaac Sim heightfields) in order to construct the anisotropic cost. On a real robot this information would need to come from onboard sensing and estimation, which will be noisy and sometimes incomplete. Extending FiRL to work with learned or filtered estimates of slope and friction, rather than direct access to simulator geometry, is a natural next step.

**Scaling and sample efficiency.**   On the MuJoCo tasks with 3–6 degrees of freedom, FiRL reaches steady performance in roughly $5\,\mathrm{M}$ environment steps per seed. The Isaac Sim quadruped tasks require significantly more data (around $3\,\mathrm{M}$ steps per task and seed in our current setup), and preliminary experiments on even higher dimensional models suggest that variance in the CVaR critic grows with dimensionality. More aggressive variance reduction, alternative distributional critics for CVaR, or better reuse of data across policies may be needed to scale FiRL to very large robots and rich perception stacks.

**Evaluation domains and hardware validation.**   Although we moved beyond flat MuJoCo benchmarks and included three Isaac Sim quadruped tasks (ramp, stairs, and beam crossing), our evaluation is still limited to controlled locomotion settings with known terrain and no explicit perception module. We do not yet include full end to end pipelines with vision, latency, and on board compute, and we do not have real robot experiments in this paper. Testing FiRL on hardware, and in scenarios where both terrain and disturbances (such as wind or deformable ground) are partially observed, remains an important direction for future work.

