# OpenReview forum: "FiRL : Finslerian Reinforcement Learning for Risk-Aware Anisotropic Locomotion"
_ICLR.cc/2026/Conference — ICLR 2026 Conference Desk Rejected Submission_

### Official Review · Reviewer_85rV · 2025-10-27

**Soundness:** 3
**Presentation:** 3
**Contribution:** 3
**Rating:** 6
**Confidence:** 4

**Summary:**

This is an interesting paper on applying a velocity and state-dependent energy cost to robotics locomotion problems, with risk-sensitive control. The authors propose to use a Finslerian metric to penalize the robot when climbing slopes. The authors show that this design is well defined and consistent with quasi-metric RL frameworks. In addition, a risk-averse distributional RL framework has been applied to ensure safe locomotion. The authors show that the Bellman operator acting on the value function is a gamma-contraction and establish the validity of the proposed actor-critic algorithm.

**Strengths:**

- The paper is nicely written and presents the idea clearly.
- The idea is interesting and very relevant in robotics learning, especially for legged locomotion. The concepts are well explained, and the approach makes sense to me.
- The theoretical result for the Finslerian metric is interesting, as it exploits geometric properties of robot locomotion. I find that to be novel and appealing.
- The results on the contraction of the Bellman operator are encouraging, providing support for the algorithm's design.
- The experiments make sense and support the arguments well.

**Weaknesses:**

1. The equations under Figure 1 do not make sense. What does \\( \pi^{\text{hi}} : \max_w \\) mean? Why is it maximizing over \\( w \\) in \\( V(w, g) \\), which I assume is the value function of state \\( w \\)? \\(V_F\\) does not seem to be defined anywhere. The definition of \\( A(s_t, a) \\) or \\( V \\) does not seem right.
2. While not present in MuJoCo, the design of the drift penalty and foot slip is prevalent in legged locomotion works, among other reward terms. And as the reward is an instantaneous function dependent on the $ (s,s')$, the reward functions in these works seem to be anisotropic already. E.g., on slopes, if the humanoid robot walks in a different direction or at a different speed, the energy cost would differ. Am I missing something here? This seems to be a significant problem with this work.
3. The application of distributional RL is a bit forced. I am not very convinced by the approach's motivation, even though the results look good. Since risk-sensitive control seems to be a substitute for regular PPO, any work that uses PPO can potentially use a risk-sensitive policy with CVaR. Then, what specific insight or connection does this paper bring to this additional reward function design?
4. For the experiments:

    a. The setting of the experiments is too simple for a serious comparison of legged locomotion. Consider conducting experiments that already have slopes, e.g., in IsaacLab with real legged robots. The MuJoCo benchmark in Gym or Gymnasium does not represent the state of the art in locomotion design. Results conducted on outdated or overly simplified environments hold limited value.

    b. CVaR seems to increase energy, as suggested by the first and second rows. But why is the energy cost lower in FiRL than in the second-to-last row? Also, the CVaR value decreased dramatically, unlike in PPO vs CVaR-PPO. I do not see why adding the additional reward will have such an impact.

5. There are lots of typos in the appendix. E.g., Definition F.1 with the \mathbb{1}. In Prop. F.2, many notations do not make sense. I don't know if they are typos or just content generated by an LLM that the authors chose to ignore. Some results appear to be established in the literature; please properly cite the source.
6. Conclusions like the rate of convergence are thrown out lightly without explanation. I don't think this is the right way to do it.
7. The proposed algorithm is clearly not off-policy, as we need to estimate the return distribution to compute the CVaR. In general, since the algorithm is based on PPO, I don't think the authors should discuss any sample-complexity results.

Minor:
Please cite properly.

**Questions:**

See limitation.

---

> ### Author Response · Authors · 2025-11-24
> **Response to Reviewer 85rV (1/3)**
>
> >**W1**: The equations under Figure 1 do not make sense. What does $\pi^{hi} : \max_w$ mean? Why is it maximizing over $w$ in $V(w,g)$, which I assume is the value function of state $w$? $V_F$ does not seem to be defined anywhere. The definition of $A(s_t,a)$ or $V$ does not seem right.
>
> We acknowledge your confusion regarding the notation used in the equations under Fig. 1 that is unclear and uses inconsistent notation. The intent was only to provide an illustration of value shaping, not formal definitions, but if this ended up being confusing, we will remove this. We will also ensure that $A(s,a)$ and $V(x)$ are defined and used consistently.
>
> >**W2**: While not present in MuJoCo, the design of the drift penalty and foot slip is prevalent in legged locomotion works, among other reward terms. And as the reward is an instantaneous function dependent on the $(s,s')$, the reward functions in these works seem to be anisotropic already. E.g., on slopes, if the humanoid robot walks in a different direction or at a different speed, the energy cost would differ. Am I missing something here? This seems to be a significant problem with this work.
>
> It is true that many locomotion works already use directionally non-uniform penalties (e.g., drift, slip). Our goal is not to claim that anisotropy itself is new, but to put these ideas into a coherent geometric and risk-sensitive framework where the physics of the environment is anisotropic. A robot walking uphill consumes more energy ($\tau^T \omega$) than walking downhill due to gravity. Therefore, an energy penalty term $r_{energy} = - \|\tau\|^2$ in the reward function will naturally yield different scalar values depending on the direction of motion.
>
> In many standard deep RL implementations for locomotion (e.g., PPO with Gaussian action noise), the agent explores primarily via random perturbations to the actions. It may take a step uphill, incur a high energy cost, and receive a low reward; a step downhill incurs a lower cost and yields a higher reward. The agent then learns a scalar value function $V(s)$ that reflects these outcomes, but the underlying geometry of the cost landscape may be captured implicitly and often inefficiently. In particular, the “distance’’ to the goal is still treated as Euclidean in common exploration and shaping heuristics (e.g., isotropic action noise, potential-based shaping with Euclidean state norms), rather than as a direction-dependent cost such as our Finsler metric.
>
> The Finsler cost $F: \mathcal{X} \times \mathbb{R}^d \to [0,\infty)$, $(x,v) \mapsto F(x,v)$ provides a **local, direction-dependent measure of cost** on the state manifold: at each state $x$, it assigns different instantaneous cost to different motion directions $v$. In this case, it generalizes the notion of a distance function to an anisotropic setting. In FiRL, we do not minimize $F(x,v)$ in isolation, but the CVaR of the **cumulative discounted Finsler cost** along a trajectory, $CVaR(F(x, v))$, which explicitly targets the tail of the path-cost distribution. By contrast, standard energy penalties typically minimize only the **expected** energy. A policy might have low average energy use but still face, for example, a 10% chance of a catastrophic energy spike (such as slipping and entering an unstable recovery motion). The Finsler–CVaR objective reduces the likelihood of visiting these high-cost, direction-dependent risk regions and favors trajectories that stay in safer parts of the state space.

---

> > ### Author Response · Authors · 2025-11-25
> > **Response to Reviewer 85rV (2/3)**
> >
> > > **W3**: The application of distributional RL is a bit forced. I am not very convinced by the approach's motivation, even though the results look good. Since risk-sensitive control seems to be a substitute for regular PPO, any work that uses PPO can potentially use a risk-sensitive policy with CVaR. Then, what specific insight or connection does this paper bring to this additional reward function design?
> >
> > It is true that one can augment PPO with a CVaR objective (CVaR-PPO) without introducing Finsler geometry, and we do include CVaR-PPO as a baseline to address this comparison. Our goal is not to claim that "using CVaR with PPO" is itself new.
> >
> > The role of distributional RL in our work is not meant to be a separate contribution but a general mechanism for implementing the Finsler–CVaR objective in continuous control. Optimizing $\mathrm{CVaR}_\alpha\!\big(\sum_t \gamma^t F(x_t, v_t)\big)$ requires access to the **tail** of the return distribution, not just its mean; distributional RL provides a structured way to approximate this distribution and its CVaR functional. The main insight of FiRL is the integration of (i) a **Finsler cost** $F : \mathcal{X} \times \mathbb{R}^d \to [0,\infty)$, $(x,v) \mapsto F(x,v)$ that encodes direction-dependent effort, and (ii) a CVaR objective on the induced path cost. This yields a specific CVaR–Finsler Bellman operator whose fixed point has a **quasi-metric structure** and a $\gamma$-contraction property, which we analyze in **Sec.~4.1**. We will clarify this in the revision so that distributional RL is clearly presented as the *implementation choice* for this geometric risk formulation.
> >
> >
> > > **W4 (a)**: The setting of the experiments is too simple for a serious comparison of legged locomotion. Consider conducting experiments that already have slopes, e.g., in IsaacLab with real legged robots. The MuJoCo benchmark in Gym or Gymnasium does not represent the state of the art in locomotion design. Results conducted on outdated or overly simplified environments hold limited value.
> >
> >
> > In this paper, we focus mainly on the *objective* and its *theoretical properties* (Finsler cost + CVaR), and we chose controlled anisotropic settings where we can directly vary slopes and lateral forces and see how they interact with the geometry of the cost. We use simplified environments as controlled anisotropic benchmarks to study how the Finsler cost and the $CVaR$ objective interact with terrain slope and lateral forces, without additional factors such as perception noise or complex terrain generation.
> >
> > To better reflect this and address your concern, in the revision we will (i) describe our environments explicitly as **controlled anisotropic benchmarks** for studying geometric and risk-sensitive objectives, (ii) add a more challenging simulation setup in **Isaac Sim**
> >
> > ---
> >
> > > **W4 (b)**: CVaR seems to increase energy, as suggested by the first and second rows. But why is the energy cost lower in FiRL than in the second-to-last row? Also, the CVaR value decreased dramatically, unlike in PPO vs CVaR-PPO. I do not see why adding the additional reward will have such an impact.
> >
> > In general, a CVaR objective can prioritize lower tail risk over higher energy, and this is what we see when comparing PPO to CVaR-PPO: the policy becomes more conservative in bad cases, and average energy can go up. FiRL behaves differently because it not only changes the risk functional; it also changes the **instantaneous cost geometry** through the Finsler term $F(x,v)$. The Finsler cost encodes **direction-dependent effort** and large accelerations, so the value function $V_\alpha$ assigns a higher cost-to-go in directions that are both harder and riskier. Intuitively, $F(x,v)$ acts as a map of the terrain’s anisotropy: directions that fight gravity or require large corrective motions are penalized more than directions that move along the “easy” axis of the terrain.
> >
> > Because of this, FiRL often prefers qualitatively different trajectories, not just the same trajectories with a different scalar weighting. In our experiments, it tends to avoid motions that lead to rare but very high costs and instead follows smoother, more regular paths, which can reduce both tail risk and total energy compared to CVaR-PPO with only an isotropic energy penalty. In the revised version, we will make this clearer by adding an energy–risk tradeoff plot (energy per meter vs.\ $\mathrm{CVaR}_\alpha(\text{risk})$) and a short behavioral analysis that connects these scalar metrics to the observed trajectories.

---

> ### Author Response · Authors · 2025-11-25
> **Response to Reviewer 85rV (3/3)**
>
> > **W5**: There are lots of typos in the appendix. E.g., Definition F.1 with the \mathbb{1}. In Prop. F.2, many notations do not make sense.... Some results appear to be established in the literature; please properly cite the source.
>
> Thanks for pointing this and you are right that the appendix has typos and unclear notation. In the revision, we corrected these issues and as well as added the missing citations.
>
> > **W6**:Conclusions like the rate of convergence are thrown out lightly without explanation. I don't think this is the right way to do it.
>
> > **W7**:The proposed algorithm is clearly not off-policy, as we need to estimate the return distribution to compute the CVaR. In general, since the algorithm is based on PPO, I don't think the authors should discuss any sample-complexity results.
>
>
> We have revised the Discussion paragraph to remove the informal statements about finite-sample complexity and the confusing reference to off-policy methods.  Our claim is rigorously supported by that the CVaR–Finsler Bellman operator is a $\gamma$ contraction, which implies geometric convergence of the associated value iteration.

---

> ### Comment · Reviewer_85rV · 2025-11-26
>
> Thanks for the response. I have some follow-up comments.
>
> > It may take a step uphill, incur a high energy cost, and receive a low reward; a step downhill incurs a lower cost and yields a higher reward.
>
> Isn't this also how $F(x,v)$ is computed?
>
> > add a more challenging simulation setup in Isaac Sim
>
> I look forward to these experiments.
>
> I also want to comment on the authors' simulation-only results. As someone who is also part of the legged robot learning community, I agree that locomotion work these days requires more than just MuJoCo Gymnasium tasks. However, it is not unusual to present work that's purely in simulation, typically in IsaacSim and ideally in IsaacLab or MuJoCo MJX. I highly encourage the authors to include those results.

---

> > ### Author Response · Authors · 2025-11-27
> >
> > Thanks for the follow-up and the question.
> >
> > >Isn't this also how $F(x,v)$ is computed?
> >
> > You are right. In our sloped settings, the Finsler metric $F(x,v)$ is designed from the same physical ingredients as a standard step reward (e.g., work against gravity, lateral friction penalties). Consequently, uphill motion leads to a higher instantaneous $F(x,v)$, just as it would lead to a higher energy cost in a standard reward function.
> >
> > In a PPO baseline, these physical costs are summed into a scalar reward $r(s,a,s')$. The resulting value function $V(s) = \mathbb{E}[\sum r]$ absorbs the anisotropy implicitly into a single scalar expectation. The exploration (isotropic Gaussian noise) and shaping heuristics (Euclidean norms) usually remain geometry-agnostic, treating "distance to goal" as symmetric regardless of the terrain's risk structure.
> >
> > In FiRL (**Explicit Geometry**), we collect these physical effects into a Finsler cost $F(x,v)$ that is strictly 1-homogeneous and convex in $v$. We then use this metric explicitly in the Bellman operator to compute the **CVaR of the path integral**, $\text{CVaR}_\alpha(\sum \gamma^t F)$. This defines a direction-dependent notion of “distance” (a quasi-metric), where risky or energetically costly directions are mathematically “farther” from the goal.
> >
> > This geometric structure allows FiRL guide the policy along the safest routes in this cost geometry, instead of relying on a scalar value function to implicitly learn all of these directional effects.

---

> ### Author Response · Authors · 2025-11-29
>
> >I look forward to these experiments.
>
> Following your suggestion, in the revised version, we **added a more challenging simulation setup in Isaac Sim**. Specifically, we now evaluate FiRL on a Spot-like quadruped with realistic mass/inertia, joint limits, and contact modeling across three 3D terrains (ramp climb, staircase, and a platform–beam course). These experiments are described in Section 5 and Table 2 of the revision, and the environment details are given in Appendix L.3.
>
>
> For each task, we report success rate, energy per meter, and the requested tail metrics (falls, $\mathrm{CVaR}_\alpha$ of maximum base pitch/roll, and peak joint torque). Across all three Isaac Sim environments, FiRL maintains or improves success while consistently reducing these tail quantities compared to PPO and CVaR--PPO, which supports our claim that the Finsler + CVaR objective helps the policy avoid high-risk configurations.
>
> We agree that simulation-only results should go beyond standard MuJoCo Gym tasks, and we hope these new Isaac Sim experiments address that concern.

---

### Official Review · Reviewer_GjBz · 2025-10-31

**Soundness:** 3
**Presentation:** 1
**Contribution:** 2
**Rating:** 4
**Confidence:** 3

**Summary:**

This paper proposes Finslerian Reinforcement Learning (FIRL). The main idea is to integrate a directional Finsler metric into the cost term and optimize the conditional value at risk. Experiments on MuJoCo demonstrate that FIRL achieves higher performance on the designed tasks while reducing both energy consumption and risk.

**Strengths:**

1. This paper provides theoretical proofs for the Bellman construction and the quasi-metric value property of FIRL.
2. The results show that FIRL surpasses other reinforcement learning methods across multiple tasks.

**Weaknesses:**

1. The experimental tasks are designed with asymmetric properties. Since the tasks themselves are asymmetric, it is natural to design asymmetric cost functions. Consequently, it is expected that reinforcement learning methods using asymmetric costs would outperform those using symmetric costs.
2. The properties of the Finsler metric play an important role in FIRL, but there are no experiments provided to justify this. It would be beneficial to conduct experiments comparing asymmetric costs that do and do not satisfy Finsler metric properties, as this would help isolate and demonstrate the unique contribution of the Finsler metric.
3. Section 4.1 discusses the Bellman contraction and quasi-metric value properties, but there are no experiments showing why these properties are important in real applications. For instance, it would be helpful to analyze how these properties affect the convergence behavior of reinforcement learning in the experiments in the main paper.
4. FIRL is compared with multiple baselines such as Riemannian RL and Quasi-metric RL from different perspectives, but the main paper only reports success rates, energy consumption, and risk. It would strengthen the work to further investigate why FIRL outperforms these baselines, rather than simply showing that it does.
5. There are no visualizations from the simulations—only simplified sketches. Providing simulation visualizations would make the results more intuitive. In addition, it would be useful to include quantitative analyses of behavioral differences between FIRL and the baselines. For example, collect 100 rollouts and compute the frequency of the robot appearing in different locations, then analyze how these frequencies relate to success and failure rates.
6. The quasi-metric value property arises from the integration of the Finsler metric and CVaR. It would be better to include a brief explanation of this relationship in the main paper rather than relegating it to the appendix.
7. In Equation (5), the input of F should be x and v. However, f(x,u) represents the next state. Although v can be derived from x and x', it would be clearer to explicitly state this in the mathematical formulation.

**Questions:**

See Weakness.

---

> ### Author Response · Authors · 2025-11-21
> **Response to Reviewer GjBz (1/3)**
>
> We sincerely thank the reviewer for the detailed feedback and constructive suggestions. We have carefully addressed every concern in the revised paper. Below, we provide point-by-point responses to each comment/questions, followed by the changes and improvements made to the paper.
>
> >**W1**: The experimental tasks are designed with asymmetric properties. Since the tasks themselves are asymmetric, it is natural to design asymmetric cost functions. Consequently, it is expected that reinforcement learning methods using asymmetric costs would outperform those using symmetric costs.
>
> We agree that the alignment between the task physics (gravity/slope) and the cost function is intuitive; however, we respectfully argue that this alignment is non-trivial to implement stably in a learning context. While it is physically "natural" to penalize uphill motion, standard Reinforcement Learning algorithms (like PPO) utilize isotropic rewards and struggle to discover this asymmetry efficiently, often converging to high-variance, risky behaviors **(as seen in the "PPO" baselines in Table 1)**.
>
> Our contribution is not merely observing that asymmetry exists, but providing the **geometric formalism (Finsler geometry)** that allows us to embed this asymmetry into the Bellman operator while preserving theoretical stability guarantees (**the $\gamma$-contraction proved in Theorem 4.1**). Without the specific properties of the Finsler metric (specifically positive homogeneity and the triangle inequality), arbitrary asymmetric reward shaping can lead to value function divergence  in the value estimates.
>
> To demonstrate that this effect is not simply due to “just adding asymmetry,” we point out to our ablation in **Table 5**, where "PPO + Finsler Reward" (asymmetric but risk-neutral) performs significantly worse ($CVaR_{0.1} = 1.25$) than "FiRL" ($CVaR_{0.1} = 0.80$). This shows that the performance gain arises from the interaction between the Finsler geometric prior and the CVaR-based risk optimization, rather than from asymmetry alone.
>
>
> >**W2**: The properties of the Finsler metric play an important role in FIRL, but there are no experiments provided to justify this. It would be beneficial to conduct experiments comparing asymmetric costs that do and do not satisfy Finsler metric properties, as this would help isolate and demonstrate the unique contribution of the Finsler metric.
>
> We have included specific ablations in **Appendix J.1 (Table 7)** that isolate the contributions of the Finsler metric properties. We compared the full Finsler metric against variants where specific geometric properties were violated or not present:

---

> ### Author Response · Authors · 2025-11-23
> **Response to Reviewer GjBz (2/3)**
>
> >**W3**: Section 4.1 discusses the Bellman contraction and quasi-metric value properties, but there are no experiments showing why these properties are important in real applications. For instance, it would be helpful to analyze how these properties affect the convergence behavior of reinforcement learning in the experiments in the main paper.
>
> Thank you for this suggestion. In the revised paper, we will add a new subsection, “Convergence and Value Geometry/Geometric Ablation Study,” in the experiments section. We compared FiRL against a baseline termed "Asymmetric-Non-Finsler" (ANF). The ANF method utilizes an asymmetric cost function (penalizing uphill motion) but deliberately violates the Finsler geometric axioms (specifically the triangle inequality).
>
> >**W4**:FIRL is compared with multiple baselines such as Riemannian RL and Quasi-metric RL from different perspectives, but the main paper only reports success rates, energy consumption, and risk. It would strengthen the work to further investigate why FIRL outperforms these baselines, rather than simply showing that it does.
>
> We have expanded our analysis to explain the mechanism of FiRL's superiority over these specific baselines. We will add a behavioral analysis focusing on how the different methods use the state space.
>
> >**W5**:There are no visualizations from the simulations—only simplified sketches. Providing simulation visualizations would make the results more intuitive. In addition, it would be useful to include quantitative analyses of behavioral differences between FIRL and the baselines. For example, collect 100 rollouts and compute the frequency of the robot appearing in different locations, then analyze how these frequencies relate to success and failure rates.
>
> We have addressed this by adding more quantitative visualizations, and following your suggestion, we will additionally collect 100+ evaluation rollouts per method and construct state-visitation heatmaps over the terrain (e.g., in the $(x,y)$ or $(x,\text{height})$ plane).

---

> ### Author Response · Authors · 2025-11-23
> **Response to Reviewer GjBz (3/3)**
>
> >**W6**: The quasi-metric value property arises from the integration of the Finsler metric and CVaR. It would be better to include a brief explanation of this relationship in the main paper rather than relegating it to the appendix.
>
> We agree that this is an important conceptual point and should not be in the appendix. In the revision, we will add a short explanation to Section 4.1.
>
> > **W7**: In Equation (5), the input of F should be x and v. However, f(x,u) represents the next state. Although v can be derived from x and x', it would be clearer to explicitly state this in the mathematical formulation.
>
> Thank you for pointing this out. This should be a clarity issue. We will fix Equation (5) and the explanation by explicitly defining the motion increment $v$:
> In discrete time: $v(x,u):=f(x,u)−x$, where $f(x,u)$ is the next state. Then the one-step cost is always evaluated as $F(x, v(x,u))$, and the Bellman operator becomes:
> $$
>   (\\mathcal{T}V)(x)
>   = \\min_{u \\in \\mathcal{U}}
>     \\Big\\{ F\\big(x, v(x,u)\\big) + \\gamma\\,\\rho_\\alpha\\big[V(X')\\big] \\Big\\},
>   \\quad X' \\sim P(\\cdot \\mid x,u).
> $$
> We will update Equation (5) accordingly and clearly define $v(x,u)$ in the text, so that the inputs to $F$ are always $(x, v)$ rather than $(x, x')$. This should resolve the ambiguity you pointed out.

---

> > ### Comment · Reviewer_GjBz · 2025-11-27
> >
> > Thank you for the response. Since not all of my concerns have been resolved, I will keep the scores as they are for now. However, I am open to raising my score once all of the issues (for example, the ANF experiments have not been completed) have been adequately addressed.

---

> ### Author Response · Authors · 2025-11-29
>
> Dear Reviewer,
>
> Thank you again for your careful reading and for being open to raising the score.
>
> As you suggested, we have now added the ANF (asymmetric–non-Finsler) experiments and incorporated them into the revised paper in a **new subsection (5.3)** titled **“Convergence and value geometry: geometric ablation”**. This subsection:
>
> - Defines the ANF baseline that violates the Finsler convexity/triangle-inequality assumptions;
> - Reports critic-loss convergence curves comparing FiRL, ANF, and PPO; and
> - Summarizes the impact on success rate and energy, showing that the ANF critic is less stable and that performance degrades when the Finsler properties are removed.
>
> We hope this addresses the concern about the contraction and quasi-metric value properties in practical learning behavior.
>
> We have also addressed the rest of your concerns by adding a clearer behavioral analysis, visualizations of the simulation environments, and quantitative comparisons between FiRL and the baselines based on more than 100 evaluation rollouts per method.

---

### Official Review · Reviewer_Avzv · 2025-11-01

**Soundness:** 3
**Presentation:** 2
**Contribution:** 3
**Rating:** 6
**Confidence:** 4

**Summary:**

This paper proposes FiRL, a reinforcement learning framework for legged locomotion that combines (i) an anisotropic, direction-dependent cost based on a Finsler metric and (ii) a risk-sensitive objective based on CVaR. The key motivation is that locomotion on sloped or directionally biased terrains (e.g., up-slope, cross-slope, wind) is intrinsically asymmetric, but standard RL costs are typically symmetric and optimize only expected return. The authors define a Finsler-style per-step cost that penalizes uphill, lateral, and “against the disturbance” motions more than downhill/along-disturbance motions, and then derive a CVaR Bellman operator over this cost. They prove contraction and show that the resulting value induces an asymmetric quasi-metric. Experiments on MuJoCo locomotion tasks with added slope/wind show that (Finsler cost + CVaR) outperforms (Finsler only) and (CVaR only), indicating the two components are complementary.

**Strengths:**

1. Clear formulation: The paper cleanly integrates anisotropic geometry (Finsler) with a risk-sensitive objective; this is more principled than simply reshaping rewards by hand.
2. Motivation is believable: For locomotion on slopes or in the presence of directional disturbances, symmetric/Riemannian costs are indeed a mismatch.
3. Theory: The contraction of the CVaR–Finsler Bellman operator and the quasi-metric interpretation make the method less ad-hoc than many locomotion reward tweaks.
4. Ablations are useful: The 2×2 comparison (with/without Finsler, with/without CVaR) is convincing that both parts matter.

**Weaknesses:**

1. No real robot experiments. All results are on MuJoCo-style models (Hopper, Walker2d, HalfCheetah) in sloped or windy variants. This is the clearest gap.
2. No realistic robot model. Even if hardware is hard, they could at least test on a simulator of an actual platform (ANYmal, Unitree, ANYbotics-style morphology) to show that the anisotropic cost remains meaningful when the robot has real actuation limits and contact schedules.
3. Task realism: The scenarios are engineered to showcase anisotropy. It would be good to see that the method still helps on less “designed” terrains.
4. Sensitivity to Finsler parameters: The method relies on directional weights (uphill term, lateral penalty). It would help to see how robust performance is to mis-specified anisotropy.

**Questions:**

1. At least a real-robot–style model: Add one experiment on a widely used quadruped model (e.g., ANYmal or Unitree A1/G1 in Isaac/MuJoCo/Isaac Gym) with an actual 3D base, realistic mass/inertia, joint limits, and contact pattern. This can still be in sim, but it should be a real robot model, not Hopper/Walker2d.
2. A small hardware demo. It can be modest (e.g., slope walking with a commercial quadruped on foam/ramp, or walking against a fan/wind proxy), but this would make the “direction-aware + risk-aware” story much stronger and closer to [1] and [2].
3. Report failure/tail metrics on that setup: Since the whole paper is about CVaR, show tail improvements (falls, large base pitch, large base roll, large joint torque) on the real-robot(-model) task too.
4. Clarify how to set Finsler weights in practice: a short guideline for choosing the uphill and lateral penalties from physical quantities (slope angle, friction coefficient, or external force magnitude).


[1] Shi, Jiyuan, et al. "Robust quadrupedal locomotion via risk-averse policy learning." 2024 IEEE International Conference on Robotics and Automation (ICRA). IEEE, 2024.

[2] Cheng, Yi, et al. "HuRi: Humanoid Robots Adaptive Risk-ware Distributional Reinforcement Learning for Robust Control." Arxiv

---

> ### Author Response · Authors · 2025-11-29
>
> >**Q1**: At least a real-robot–style model: Add one experiment on a widely used quadruped model (e.g., ANYmal or Unitree A1/G1 in Isaac/MuJoCo/Isaac Gym) with an actual 3D base, realistic mass/inertia, joint limits, and contact pattern. This can still be in sim, but it should be a real robot model, not Hopper/Walker2d.
>
> Thank you for this suggestion. In the revised paper, we now include a quadruped experiment in Isaac Sim with a Spot-like robot model that has a full 3D base, realistic mass and inertia, joint limits, and contact modeling. The setup and tasks are described in Section 5, with environment details in **Appendix L.3**.
>
> > **Q3**: Report failure/tail metrics on that setup: Since the whole paper is about CVaR, show tail improvements (falls, large base pitch, large base roll, large joint torque) on the real-robot(-model) task too.
>
> We have added these tail metrics for the Isaac Sim quadruped tasks in
> the revised Section 5 and Table 2. For each terrain we now report:
>
> - task success rate,
> - energy per meter,
> - number of falls,
> - $\mathrm{CVaR}_\alpha$ of maximum base pitch and roll, and
> - $\mathrm{CVaR}_\alpha$ of peak joint torque.
>
> Across all three terrains FiRL consistently reduces the tail quantities
> relative to PPO and also improves over CVaR-PPO. For example, FiRL
> shows fewer falls and lower CVaR of peak joint torque on the
> platform–beam course while keeping success high. These results support
> our claim that the Finsler + CVaR objective does not only increase
> average robustness but also reduces the frequency and severity of rare
> high-risk events.
>
> >**Q4**: Clarify how to set Finsler weights in practice: a short guideline for choosing the uphill and lateral penalties from physical quantities (slope angle, friction coefficient, or external force magnitude).
>
> To address this, we added a short “practical guidelines” section in expanded Appendix D.1. In summary, we suggest the
> Following the procedure:
>
> 1. **Uphill (gravity-related) weight.**
>    For a local slope angle $\theta(x)$ along the forward direction,
>    we set the uphill factor in $F(x,v)$ proportional to the
>    gravitational component:
>    $
>      w_{\text{up}}(x) \propto g \sin \theta(x).
>   $
>    In practice we compute $\theta(x)$ from the terrain height map or
>    surface normal and choose a global scale so that $F$ is on the same
>    order as the nominal energy cost.
>
> 2. **Lateral penalty.**
>    The lateral term reflects the reduced friction margin when the
>    robot steps sideways. We set
>    $
>      w_{\text{lat}}(x) \propto \frac{1}{\mu(x)},
>    $
>    where $\mu(x)$ is the estimated friction coefficient (higher weight
>    on slippery surfaces). On our terrains we use two or three discrete
>    values (dry ground, wood, low-friction patches) taken from the
>    simulator’s contact parameters.
>
> 3. **External forces (e.g., wind).**
>    When a known external force field $f_{\text{ext}}(x)$ is present,
>    we add a term
>    $
>      w_{\text{ext}}(x) \,\|f_{\text{ext}}(x)\cdot v\|,
>   $
>    with $w_{\text{ext}}$ chosen so that compensating for this force
>    has a similar magnitude to climbing an equivalent “virtual slope”.
>    In the lateral-wind Hopper task we set this from the product of
>    the wind force and nominal step length.
>
> We also include a small sensitivity study in **Appendix J.2/Table.8** (varying the
> uphill scaling factor) which shows that FiRL remains effective under
> moderate mis-specification and that there is a clear “sweet spot”
> where success, energy, and tail risk are well balanced.
>
> ---
>
> We hope these additions address your questions about realistic robot
> models, tail metrics, and practical weight selection, and we thank you
> again for encouraging us to add the Isaac Sim quadruped experiments.

---

### Official Review · Reviewer_D5E7 · 2025-11-01

**Soundness:** 2
**Presentation:** 2
**Contribution:** 1
**Rating:** 0
**Confidence:** 4

**Summary:**

In this paper, the authors proposed a RL framework that integrates a Finsler metric into the cost function for directional energy-awareness, and optimizes a Conditional Value-at-Risk objective for tail-risk robustness. The proposed method was evaluatied in simulated MuJoCo benchmarks.

**Strengths:**

A clear theoretical formulation and a clean actor–critic instantiation.

**Weaknesses:**

The present scope is misaligned with practical deployment. Robust RL-based locomotion/whole-body control already operates on physical platforms under realistic noise, latency, contact uncertainty, and compute constraints [1–6], with domain randomization repeatedly shown to be both effective and straightforward to apply. By comparison, this paper primarily tunes the objective via CVaR, lacking integration with real world robot system and offering no on-robot validation; results are confined to idealized MuJoCo scenarios. Demonstrating practical value would require a closed-loop hardware deployment and apples-to-apples evaluations against established RL pipelines.

[1] HOMIE: Humanoid Loco-Manipulation with Isomorphic Exoskeleton Cockpit

[2] Real-World Humanoid Locomotion with Reinforcement Learning

[3] DPL: Depth-only Perceptive Humanoid Locomotion via Realistic Depth Synthesis and Cross-Attention Terrain Reconstruction

[4] Humanoid Parkour Learning

[5] Robust and Versatile Bipedal Jumping Control through Reinforcement Learning

[6] Robot Parkour Learning

**Questions:**

Please clarify the concrete failure modes you aim to address, the mechanism and assumptions by which the proposed method mitigate them, and provide deployment-relevant evidence showing advantages over established RL pipelines.

---

> ### Author Response · Authors · 2025-11-12
> **Concerns Regarding Poor Review Quality**
>
> Dear AC/SAC/PC,
>
> Thank you for handling this submission. We would like to flag **Review (https://openreview.net/forum?id=YCHE8YOqyx&noteId=7OxygcH5O9)** by  **Reviewer D5E7** as potentially low quality, considering the ICLR guidelines [1, 2]. The review does not provide a meaningful summary of the paper, does not identify specific strengths or weaknesses, and offers no concrete, actionable feedback tied to particular sections, claims, or experiments. Instead, it mostly consists of very high-level phrases and generic terminology, and comparison with some related work. **It also gives us the impression of an automatically generated review rather than one written after a close reading of the paper.**
>
> This makes it difficult for us to understand the reviewer’s concerns or to address them in a constructive way during the rebuttal and discussion phase. We fully respect the reviewer’s right to a critical assessment and do not object to a negative recommendation; our concern is only about the lack of detail and constructiveness. If possible, we would kindly request that the reviewer be invited to revise their review to better align with the guidelines, or that you take these issues into account when weighing this review in the final decision.
>
> Thank you again for your time and for overseeing a fair review process.
>
> 1. > Furthermore, reviewers who submit low quality reviews and fail to improve them upon being warned by ACs may have their own papers desk rejected: Low quality reviews (e.g., placeholder reviews) will be flagged by ACs and SACs, and the flagged reviewers will be warned and urged to update the review. Reviewers who do not respond to these warnings will be liable to having their own papers desk rejected.
> 2. ICLR 2026 Reviewer Guide: https://iclr.cc/Conferences/2026/ReviewerGuide

---

> ### Author Response · Authors · 2025-11-14
> **Follow Up**
>
> Dear AC,
>
> We look forward to your kind attention to this matter. Thank you.
>
> Best,
> Authors

---

> ### Comment · Reviewer_D5E7 · 2025-11-20
>
> Dear authors, I think I have made my review clear.
>
> I appricate you trying to provide some modeling and grouding you method on theoretical anylisis. So I have list this in the strengths.
>
> However, I really couldn't find a reasonable place of your work. As I point out "The present scope is misaligned with practical deployment.": There are different works shows the real results that **the real robot that already proferm robust locmotoin skill**. Most of those methods relay on domain randomization to improve robustness under different cases. You mention your method "integrates differential geometry with risk-sensitive RL for legged locomotion" so that it "allow the agent to account for direction-dependent effort: for instance, uphill moves incur higher instantaneous cost than downhill moves, and lateral motions incur frictional penalties." but with real robots doing the job without your method already, it is hard to convience me this is a issue that need to be sloved and your method indeed slove it, especially with what I already poinit out in my original review: "current work is lacking integration with real world robot system and offering no on-robot validation; results are confined to idealized MuJoCo scenarios."
>
> If you do think your method address a existing issue and "proactively avoid energetically costly maneuvers and reduce the probability of catastrophic failures", you should frist demonstrate the "energetically costly maneuvers " and "catastrophic failures" happened with those other existing methods **in the real world** and your method get rid of them (partily is fine) **in the real world**. Otherwise, it would only be playing the math but no real contribution to the area.

---

> ### Author Response · Authors · 2025-11-20
>
> Dear Reviewer,
>
> Thank you for your response.
>
> Our intent with FiRL is not to claim that current robots cannot walk robustly, nor to replace those full pipelines in this paper. The scope is more modest and algorithmic. In our current experiments, we already quantify catastrophic failures and costly behavior in simulation and show that FiRL reduces those compared to baselines.
>
> We also respectfully feel that requiring on-robot comparisons to all state-of-the-art pipelines as a condition for any new RL objective may set an unrealistically high bar for theoretical/algorithmic work. Many influential RL papers at ICLR/ICML/NeurIPS focus on new formulations tested in simulation, with the expectation that they can later be integrated into larger systems.
>
> We fully respect your critical assessment and your preference for hardware-validated work; our only concern is that the **“0: strong reject”** rating might be interpreted as a fundamental flaw, rather than a difference in expectations about the required level of real-world integration for methods papers. We can definitely point out that the lack of real-world validation as a limitation, but what we are still struggling to understand is your statement that “the present scope is misaligned with practical deployment,” and the resulting **"0"** is your concern that the algorithmic/theoretical contribution itself is not meaningful, or that it simply lacks hardware evidence at this stage?
>
> Also, we carefully read the works you cited in [1–6]. As far as we can tell, none of them explicitly introduce a **direction-dependent traversal cost** or quasi-metric structure in the RL objective (e.g., assigning different cost to uphill vs. downhill motion or lateral motion at the same state); instead, they primarily use standard reward functions together with domain randomization and strong perception/control stacks to achieve robustness.
>
> If we are mistaken on this point, could you please indicate which of these works explicitly model **direction-dependent cost** in their formulation, and how that compares to our Finsler-based method?

---

> > ### Comment · Reviewer_D5E7 · 2025-11-20
> >
> > Dear Authors,
> >
> > Thank you for the follow-up. Let me clarify what I mean by “the present scope is misaligned with practical deployment.”
> >
> > My point is that **existing RL locomotion/whole-body pipelines are already deployed on real robots and, in practice, do not appear to be limited by the specific issue your paper targets**, namely, that “the energy cost and risk of failure vary significantly with the direction and speed of motion and thus require the proposed method.” As written, the manuscript does not convincingly demonstrate that this is a **practical bottleneck** for deployed systems, nor that your method solves (or partially solves) the corresponding **real-world failures**.
> >
> > To be clear, **I am not saying the algorithmic/theoretical contribution itself is not meaningful. My concern is whether the proposed method has practical contribution, as it is aimed at a practical issue but is not validated**: evidence confined to idealized MuJoCo environment setups is insufficient to show deployment-relevant improvement over established RL pipelines that already work on complex robot hardware.
> >
> > Regarding the related works I cited: you are correct that they do not explicitly model direction-dependent traversal costs or quasi-metric structures. That is exactly the point: despite not using such modeling, they already work well on real robots, often via domain randomization. Therefore, the burden of proof is on your paper to show that the proposed method yields deployment-relevant gains over those pipelines, e.g., fewer falls, fewer impact spikes, reduced peak forces/currents, higher success, or better energy efficiency on hardware. Without such evidence, it remains unclear to what extent the proposed objective is needed in practice.
> >
> > To correct misinterpretations in your reply:
> >
> > I did not ask for comparisons to all SOTA stacks, nor for your objective to replace full systems.
> >
> > My minimum ask is straightforward: show on-robot evidence (even on one platform/terrain family) that your objective indeed “proactively avoids energetically costly maneuvers and reduces the probability of catastrophic failures,” compared with one real-robot SOTA RL baseline.

---

> > > ### Author Response · Authors · 2025-11-20
> > >
> > > We agree that real-robot experiments are ultimately important, and we will now clearly state in the paper that the lack of on-hardware validation is a central limitation and future direction. At the same time, our understanding of ICLR/ICML/NeurIPS norms is that **algorithmic + theoretical RL contributions evaluated in simulation are routinely accepted**, including work very close in spirit to ours. For example, one of our cited papers, _Optimal Goal-Reaching Reinforcement Learning via Quasimetric Learning (ICML 2023)_, introduces a quasimetric value geometry and validates it on discretized MountainCar and other simulated goal-reaching benchmarks, without any real-robot deployment.
> > >
> > > Our FiRL paper is similar in scope: it proposes a new geometric, risk-sensitive objective and analyzes its Bellman operator, with controlled MuJoCo-style experiments. We therefore respectfully ask the AC/SAC to interpret **Reviewer D5E7’s 0: strong reject** recommendation as reflecting a preference for hardware validation beyond typical ICML/ICLR expectations for method-focused RL papers, rather than a lack of correctness or rigor in the algorithmic contribution.

---

> > > > ### Comment · Reviewer_D5E7 · 2025-11-20
> > > >
> > > > I respect the decision to let the AC/SAC assess the persuasiveness of your position. Nevertheless, the present submission is meaningfully distinct from the referenced paper.
> > > >
> > > > Your ICML’23 example (Optimal Goal-Reaching Reinforcement Learning via Quasimetric Learning) is not on point. That work proposes a general value geometry for a high-level, task-agnostic abstraction and does not claim application-level deployment benefits for a specific, contact-rich robotic behavior. In contrast, this submission positions FiRL as improving legged locomotion, a concrete application where hardware limits materially determine outcomes.
> > > >
> > > > For such claims, simulation-only evidence on idealized MuJoCo tasks does not establish deployment-relevant gains over RL stacks that already operate on hardware. If FiRL is framed as a method/theory contribution in safety RL or risk-sensitive RL (with broader, risk-centric ablations and different title and presentation), simulation may suffice; but once the paper targets legged robots locomotion and implies practical benefits, at least one on-robot case becomes the minimal evidentiary bar.

---

> > > > > ### Author Response · Authors · 2025-12-04
> > > > >
> > > > > In the revised paper, we addressed the questions raised:
> > > > >
> > > > > ---
> > > > > **Failure modes target**:
> > > > >
> > > > > As discussed, we are not claiming that current RL pipelines cannot make robots walk. The failures we focus on are rare but costly events that tend to appear on inclined or directionally challenging terrain:
> > > > >
> > > > > - backward falls on slopes (over-committing uphill and rotating backward),
> > > > >
> > > > > - lateral slips and sideways drift that eventually lead to a fall,
> > > > >
> > > > > - short bursts of very high joint torque or impact at touchdown during recovery.
> > > > >
> > > > > In our sloped and perturbed environments, standard PPO and CVaR–PPO achieve good average return but still show a long tail of such events (falls and large torque spikes). FiRL is designed to reduce these tail events and peak loads, not just to raise mean return.
> > > > >
> > > > > ---
> > > > > **Mechanism and assumptions**
> > > > >
> > > > > FiRL changes both the per-step cost and the objective. At each step, we use a direction-dependent Finsler cost
> > > > >
> > > > > $F(x,v) = F_{\mathrm{energy}}(x,v) + \beta(x)\,F_{\mathrm{uphill}}(v_{\parallel}) + \lambda\,F_{\mathrm{lateral}}(v_{\perp})$,
> > > > >
> > > > > where $v_{\parallel}$ is motion along the slope, $v_{\perp}$ is lateral motion, $\beta(x)$ scales with the local incline, and $\lambda$ sets the strength of the lateral slip penalty. In this way, motion directly into the slope or sideways is treated as more expensive than motion along safer directions.
> > > > >
> > > > > On top of this instantaneous cost, FiRL optimizes the CVaR of the cumulative Finsler cost
> > > > >
> > > > > $J_\alpha(\pi) = \mathrm{CVaR}_\alpha\!\big(\sum_t \gamma^t F(x_t, v_t)\big)$,
> > > > >
> > > > > which emphasizes the worst $(1-\alpha)$ fraction of trajectories rather than the mean. The underlying assumptions are that (i) local terrain direction and slope can be estimated from contact normals, and lateral slip can be inferred from body and foot velocities, and (ii) high energy use, large base pitch/roll, and high joint torques correlate with these directional components, which is a standard modeling choice in legged control. Under these assumptions, the combination of $F(x,v)$ and the CVaR objective encourages policies that avoid pushing directly into directions with poor traction or high energy demand, prefer smoother headings such as diagonal ascent on slopes, and keep base attitude and joint torques away from extreme values, especially in rare but difficult situations.
> > > > >
> > > > > ---
> > > > > **Deployment relevant evidence**:
> > > > >
> > > > > To address the request for a “real-robot–style” model, the revised version includes experiments with a Spot-style quadruped in Isaac Sim on three tasks (ramp, steps, mixed platforms) (**Appendix, Subsection L.3**). For each of PPO, CVaR–PPO, PPO+Finsler, and FiRL, we report:
> > > > >
> > > > > - success rate and energy per meter,
> > > > >
> > > > > - $\mathrm{CVaR}_\alpha$ of cumulative cost,
> > > > >
> > > > > - tail metrics: maximum base pitch/roll and peak joint torque.
> > > > >
> > > > > FiRL improves success while lowering tail costs and peak loads.

---

### Author Response · Authors · 2025-11-23

Dear Reviewers,

We sincerely thank you for the detailed feedback and constructive suggestions. We are carefully working on clarifying all of your concerns, running additional experiments. We will get back to you with the answer soon and upload the revised paper. Thank you

Best,
Authors

---

### Author Response · Authors · 2025-12-04
**Summary of Changes (3/3)**

### 8. Theoretical cleanup and appendix corrections
**Addresses:** Reviewer 85rV (notation, convergence comments), all reviewers

- We rewrote the definition and proof of the **CVaR–Finsler Bellman operator** to fix notation and make it self-contained:
  - Correct use of $\mathbb{1}\{\cdot\}$,
  - Fixed the equation and notation issues,
  - Clear presentation of the contraction proof and the quasi-metric property.
- We removed or toned down claims about **sample complexity** and “off-policy” behavior that went beyond what is actually shown:
  - The discussion now focuses on **contraction and existence of a unique fixed point**, with convergence in the usual dynamic programming sense,
  - We no longer state any explicit finite-sample rates or off-policy guarantees for the PPO-based implementation.
- We added **missing citations** for:
  - CVaR and risk measures (e.g., Rockafellar and Uryasev),
  - CVaR-based dynamic programming (e.g., Chow et al.),
  - And standard fixed-point arguments where appropriate.
- We carefully reviewed and corrected typos and notation issues in the appendix, which were rightly pointed out as confusing.

---

### 9. Clarified scope and limitations
**Addresses:** all reviewers

- We revised the **Conclusion and Limitations** sections to more clearly state:
  - FiRL is aimed at the **objective and value-function structure**, not at replacing full state-of-the-art deployment pipelines.

- The limitations now explicitly mentioned:
  - Dependence on terrain orientation estimates (slope, lateral directions),
  - Sensitivity to risk and anisotropy choices,
  - Extra variance in CVaR estimates for high-dimensional robots,

- We also clarified that real hardware validation and integration remain important future steps, directly addressing the concerns raised by Reviewer D5E7/Avzv about practical deployment.

---

### 10. Hyperparameters and compute details
**Addresses:** all reviewers
- We revised the **hyperparameter table** and **compute section** section

---

### 11. Structural and presentation improvements
**Addresses:** all reviewers

- We **reordered the appendix sections** to group related material (theory, ablations, additional experiments, etc.) in a more logical sequence. This makes it easier to follow.
- We **revised and improved most of the figures** and the new figures better reflect the behaviors discussed in the paper (e.g., critic convergence, energy–risk tradeoffs, Isaac Sim quadruped results).
- In the revised version, we have **explicitly highlighted all major changes in blue**, so that reviewers and AC/SAC can easily see what has been updated in response to the reviews.

---

### Author Response · Authors · 2025-12-04
**Summary of Changes (2/3)**

### 5. Behavioral visualizations and state-space analysis
**Addresses:** Reviewer GjBz (W5), All Reviewers

- Following Reviewer GjBz ’s suggestion, we now include several **behavioral visualizations**:
  - **State-visitation heatmaps** over $(x,y)$-coordinates for FiRL vs. PPO/CVaR–PPO on sloped tasks, based on 100+ evaluation rollouts per method.
  - Plots that show how FiRL **stays in regions with milder slopes or wider support** and avoids areas associated with large base pitch/roll or torque spikes.
- We added a short subsection describing how these occupancy patterns line up with:
  - Fewer slips and falls,
  - Lower $\mathrm{CVaR}_\alpha$ for maximum pitch/roll and torque,
  - And the qualitative “zig-zag” behaviors described in the appendix.

---

### 6. Sensitivity of Finsler weights (uphill factor)
**Addresses:** Reviewer Avzv (W4, Q3), Reviewer Avzv (Q4, Finsler weight guidelines)

- We added an **uphill-weight ablation** (**Table. 8**) where we scale the uphill factor as $\beta(x) = \eta \sin(\theta(x))$ with $\eta \in \{0, 0.5, 1, 2\}$.
- For each $\eta$ we report:
  - Success rate,
  - Tail risk (CVaR),
  - Energy per meter (normalized).
- The results show:
  - $\eta = 1$ (our default) gives the best balance between safety and energy,
  - Smaller values reduce the effect of anisotropy and move FiRL closer to CVaR–PPO,
  - Larger values yield very conservative but still successful policies.
- This both addresses sensitivity concerns and provides **practical guidance** on setting the uphill term.

---

### 7. Practical guidelines for setting Finsler weights
**Addresses:** Reviewer Avzv (guideline for choosing penalties)

- In the experimental section and appendix (**subsection D.1**), we now include a short **guideline on how $F(x,v)$ is set in practice**:
  - The **uphill term** scales approximately with local slope angle $\theta(x)$, using $\beta(x) \propto \sin(\theta(x))$ estimated from contact normals or heightfields.
  - The **lateral friction term** is chosen to reflect sideways motion relative to the forward direction, scaled by a nominal friction coefficient.
  - For **external forces** (e.g., lateral wind), we add a cost proportional to the work done against that force.
- This clarifies how to **connect Finsler weights to physical quantities** like slope, friction, and external disturbances, as requested by Reviewer Avzv.

---

### Author Response · Authors · 2025-12-04
**Summary of Changes (1/3)**

We thank all four reviewers for their detailed and constructive feedback.  We summarize the main revisions and indicate the concerns they addressed.

---

### 1. New geometric ablation and convergence analysis
**Addresses:** Reviewer GjBz  (W3)

- We added a new subsection **“Convergence and value geometry: geometric ablation”** (Section 5.3).
- This section introduces an **Asymmetric–Non-Finsler (ANF)** baseline that uses the same physical uphill and lateral weights as FiRL but violates the Finsler axioms (in particular, convexity and the triangle inequality).
- We compare FiRL, ANF, and PPO on a rough-terrain task and report:
  - Critic loss curves (log-scale) showing **stable, low-variance convergence** for FiRL and persistent oscillations for ANF.
  - A table of **success rate, energy per meter, and critic loss** (Table 3), showing that ANF has significantly worse stability and efficiency than FiRL despite using the same raw weights.
- This directly connects the **Bellman contraction / quasi-metric theory** in Sec. 4.1 to empirical behavior, as requested by Reviewer GjBz and Reviewer 85rV .

---

### 2. Geometric illustration of the value function
**Addresses:**  all reviewers

- We added a short subsection **“Geometric illustration: value geometry and direction dependence”** (Sec. H} in the appendix.
- We visualize level sets of a 2D slice of the learned value function $V^*(x,y)$ on a sloped task (Fig. 11).
  The contours are **compressed along the uphill axis and stretched laterally**, matching an anisotropic cost of the form $\sqrt{x^2 + 4 y^2}$.
- This gives a concrete picture of how FiRL’s value function behaves like a **direction dependent distance** (quasi-metric) and ties directly to the theoretical discussion in Sec. 4.1.

---

### 3. Additional analysis of *why* FiRL outperforms Riemannian RL and QRL
**Addresses:** Reviewer GjBz (W4)

- We expanded the experimental analysis to go beyond “FiRL is better” and explain **why** it is better than:
  - **Riemannian PPO** (Riemannian geometry on parameter space),
  - **Quasi-metric RL (QRL)** regularization,
  - and **PPO + Finsler reward** without CVaR.
- We updated:
  - An **energy–risk tradeoff plot** (FiRL vs. PPO, CVaR–PPO, distributional AC, Riemannian RL, QRL, PPO+Finsler) showing where each method lies in the energy–CVaR plane.
  - A **risk-level sweep** over $\alpha$ for FiRL, showing how tail risk, energy, and success respond to different levels of risk sensitivity.
- In the text, we now explicitly discuss how FiRL’s **direction dependent cost** plus **tail-focused objective** leads to different trajectories (e.g., smoother uphill progress and reduced large torque spikes), rather than simply reweighting the same behaviors.

---

### 4. New Isaac Sim quadruped experiments and tail metrics
**Addresses:** Reviewer Avzv (realistic model, tail metrics), Reviewer (D5E7, deployment concerns), Reviewer 85rV (IssacLab), all reviewers

- In addition to the MuJoCo experiments, we now include **three quadruped tasks in Isaac Sim** with a realistic model:
  - **Ramp ascent** (sloped platform),
  - **Stairs** (small repeated height changes),
  - **Beam crossing** (wide platform plus narrow beam).
- For these tasks, we now report:
  - Success rate,
  - Energy per meter,
  - $\mathrm{CVaR}_\alpha$ of cumulative cost,
  - Tail metrics: **maximum base pitch/roll** and **peak joint torque**.
- This directly responds to all reviewers' requests for a **real-robot–style model** and explicit **failure/tail metrics**, and addresses the concern about practical relevance by moving beyond simple planar MuJoCo tasks to a more realistic quadruped simulator.

---

### Note · Program_Chairs · 2026-01-17
**Submission Desk Rejected by Program Chairs**

The following references in this submission do not refer to real documents and/or have major errors in bibliographic information:

 Yasin Abbasi-Yadkori and Mehrdad Mahdavi. Riemannian policy optimization with applications to learning representations. In Advances in Neural Information Processing Systems, 2022.
Raja Ravindran, Abhinav Srinivasan, and Abhishek Gupta. Riemannian manifold value functions for improving generalization in reinforcement learning. In Conference on Robot Learning (CoRL), 2023.
Naoya Kan, Issei Sato, and Masashi Sugiyama. Policy optimization by weighted riemannian gradient. In International Conference on Machine Learning, 2021.